# Printed origami thermoelectric generator achieves > 20 Wm⁻² from low-grade heat via material and process design

Nan Luo [1], Zirui Wang [1], Ajay Kumar Verma[1], Muhammad Irfan Khan [1], Leonard Franke[1,2], Jiayi Liu [1], Alexei Nefedov [3], Marc Schneider[4], Holger Geßwein[5], Erich Müller[6], Kirsten Drüppel[7], Tobias Weingaertner [8], Yolita M. Eggeler[6], Uli Lemmer[1,4] ✉ & Md Mofasser Mallick [1,2] ✉

Printing facilitates low-cost thermoelectric generators to power battery-free internet-of-things devices, wearables, and Industry 4.0 systems. However, scaling up requires printable thermoelectric materials with good mechanical properties and high performance. Here, we report a high-performance $Ag_2(Se_{1-x}S_x)_{1.05}$-based n-type printed thermoelectric film through a combination of engineering non-stoichiometric defects and sulfur substitution. An optimal sulfur substitution of 2 at. % facilitates an excellent flexibility and a power factor of~16 $\mu Wcm^{-1} K^{-2}$ at 360 K, a 65 % increase compared to a pristine $Ag_2Se$ film. A fully printed origami-thermoelectric generator produces a maximum power output $P_{max}$ of 907 $\mu W$ at a temperature difference of 80 K. A record-high power density $p_d$ of 21 W m⁻² (corresponding to 800 $\mu W\,g^{-1}$ as a weight-normalized power density) is achieved, twice that of previously reported origami-thermoelectric generators. These results highlight cost-effective manufacturing of thermoelectric generators with the capability to power next-generation autonomous electronic devices.

At present, a significant amount of primary energy used in our modern society is dissipated as waste heat[1]. Therefore, the conversion of this large amount of abandoned waste heat into useful electricity through technologies such as thermoelectric generators (TEGs) possesses significant promise[1–3]. Meanwhile, the rapidly growing Internet-of-everything (IoE) relies on billions of powered sensors and other small electronic devices requiring only micro- to milliwatts of input power[4]. In such a scenario, TEGs could become a dominating technology, provided that scalable and low-cost manufacturing solutions are achieved. Many TEGs based on bulk thermoelectric (TE) materials such as chalcogenides, Si-Ge, and skutterudite alloys with good TE

performance have been reported[5–7]. However, these TEGs have two major disadvantages: (1) the complex and expensive manufacturing process, and (2) the rigidity of the TE materials does not facilitate flexibility and conformability. Instead, printing technologies can offer low-cost production, shape-conformable manufacturing, and good flexibility of TEGs, and can be scaled up cost-efficiently[8]. Therefore, in recent times, research on printed TE materials and devices is growing to achieve scalability and cost-effectiveness[9,10]. Organic-based TE materials were initially targeted due to their mechanical flexibility and good printability[11–13]. A scalable, fully printed flexible organic TEG with a power density of 15 nW cm⁻² and an estimated weight-normalized

[1]Light Technology Institute, Karlsruhe Institute of Technology (KIT), Karlsruhe, Germany. [2]varmo UG (haftungsbeschränkt), Eggenstein-Leopoldshafen, Germany. [3]Institute of Functional Interfaces, Karlsruhe Institute of Technology (KIT), Eggenstein-Leopoldshafen, Germany. [4]Institute of Microstructure Technology, Karlsruhe Institute of Technology (KIT), Eggenstein-Leopoldshafen, Germany. [5]Institut für Angewandte Materialien (IAM-ESS), Karlsruhe Institute of Technology (KIT), Eggenstein-Leopoldshafen, Germany. [6]Laboratory for Electron Microscopy, Karlsruhe Institute of Technology (KIT), Karlsruhe, Germany. [7]Institute for Applied Geosciences, Karlsruhe Institute of Technology (KIT), Karlsruhe, Germany. [8]Institut für Angewandte Materialien (IAM-AWP), Karlsruhe Institute of Technology (KIT), Eggenstein-Leopoldshafen, Germany. ✉e-mail: uli.lemmer@kit.edu; mofasser.mallick@kit.edu

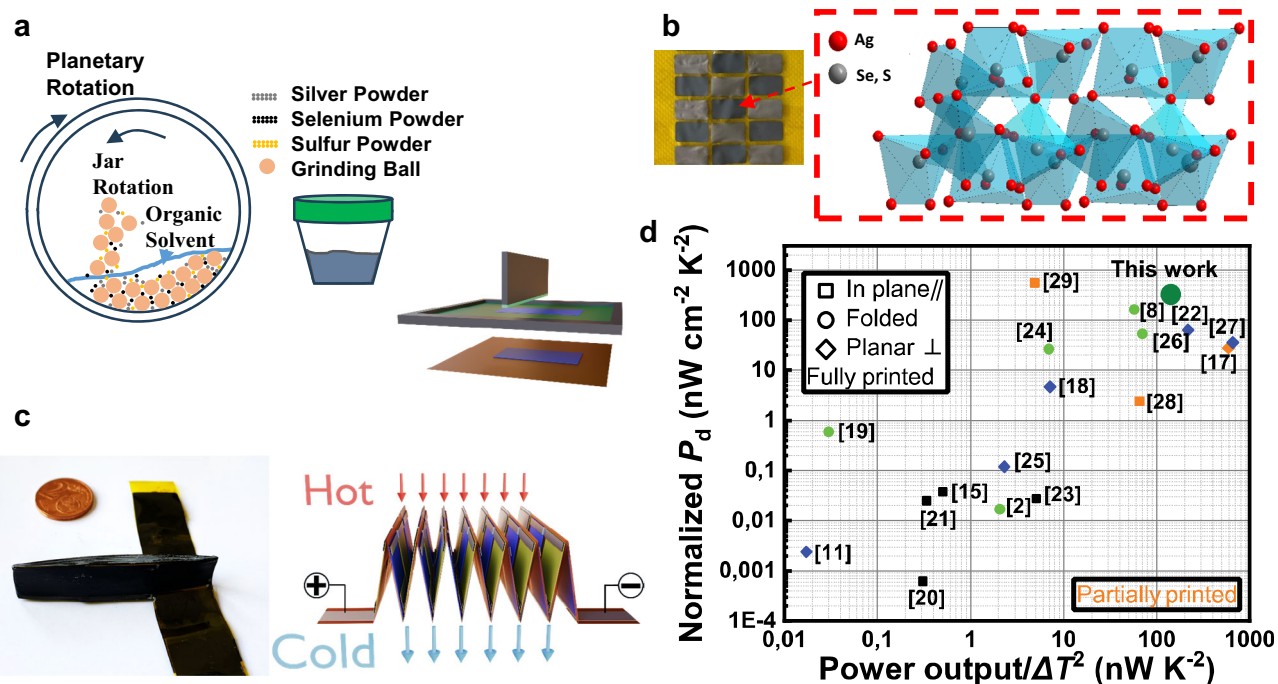

**Fig. 1 | Fabrication of origami-TEG and its performance. a** $Ag_2(Se_{1-x}S_x)_{1.05}$ ink synthesis and screen printing of the same. **b** Printed n-type $Ag_2(Se_{0.98}S_{0.02})_{1.05}$ and p-type $Bi_{0.5}Sb_{1.5}Te_3$ films. **c** Schematic of printed origami-TEG. **d** Comparison of normalized power density among recently reported partially and fully printed TEGs[2,8,11,15,18,24–35].

power output of 1 μW g⁻¹ at a temperature difference of 29.5 K has been reported[11]. The selection of materials for fully printed TEGs is a sensitive chore. Recently, the focus has shifted to printable inorganic TE materials[2,8,14]. Among them, the $Ag_2Se$-based printed TE materials have been targeted due to their high performance at room temperature and ease of processing. Yan Liu et al. reported that their fully inkjet-printed $Ag_2Se$/15% Ag composite film exhibited an electrical conductivity of 1040.2 S cm⁻¹ at room temperature, leading to a power factor of up to 8.89 μW cm⁻¹ K⁻²[15]. Kumar et al. fabricated $Ag_2Se$ films by printing an n-type $Ag_2Se$ layer onto a flexible polyimide (PI) substrate. These films exhibited impressive maximum power factors of 14 μW cm⁻¹ K⁻² at 300 K[16]. We recently reported a high-performance $Ag_2Se$-based printed n-type TE material, which achieved a high and an estimated Fig. of merit ZT of 1.03 at room temperature[14]. Based on the high TE performance of $Ag_2Se$, an origami TEG module was introduced[8]. A comparison of the recently reported normalized power densities of printed TEGs is shown in Fig. 1d. The diagram comprises in-plane as well as conventional (π-type) devices. Moreover, the possibility to fold mechanically flexible substrates allows conversion of in-plane geometries to π-type devices.

In this work, we have developed flexible non-stoichiometric $Ag_2(Se_{1-x}S_x)_{1.05}$-based screen-printed TE films (cf. Figure 1). The TE performance of the films has been enhanced by non-stoichiometric defects and sulfur (S) incorporation. The printed films were compacted by hot pressing, which further increases their flexibility and robustness. This strategy facilitates both high TE performance and flexibility of the n-type printed $Ag_2(Se_{1-x}S_x)_{1.05}$ films. The non-stoichiometry of the printed film enhances the Seebeck coefficient ($\alpha$) significantly; meanwhile, the sulfur inclusion increases the electrical conductivity ($\sigma$). As a result, a high power factor of 16 μW cm⁻¹ K⁻² is achieved at 360 K for $x = 0.02$. With this film, we fabricated and characterized a fully printed origami-TEG module (cf. Fig. 1). The origami TEG is entirely fabricated via screen printing using p-type $Bi_{0.5}Sb_{1.5}Te_3$ and the newly developed n-type $Ag_2(Se_{0.98}S_{0.02})_{1.05}$ material, followed by hot pressing, encapsulation, and folding. A high power output of around 907 μW with a power output density of 21 W m⁻² at a temperature difference ($\Delta T$) of 80 K.

## Results and discussion

### Thermal and phase analysis of printed $Ag_2(Se_{1-x}S_x)_{1.05}$ films

To evaluate phase purity and structural changes upon sulfur substitution, X-ray diffraction (XRD) was carried out on pristine non-stoichiometric $Ag_2Se_{1.05}$ and sulfur-substituted samples with nominal compositions $Ag_2(Se_{1-x}S_x)_{1.05}$ ($x = 0$, 0.02, 0.05, and 0.1), as presented in Supplementary Fig. S1 and Fig. 2. Rietveld refinement analysis of all the XRD patterns indicates that the films are crystallized in the orthorhombic phase of beta-$Ag_2Se$, in agreement with the standard JCPDS reference (No. 24–1041)[17]. The diffraction peaks can be indexed to the orthorhombic structure with space group symmetry $P2_12_12_1$, with no significant secondary phases, indicating high phase purity and crystallinity in all samples. Although the printed films are $Se_{1-x}S_x$-rich, their XRD patterns correspond to the stoichiometric beta-$Ag_2Se$ phase, consistent with a previous report[18]. To analyze the structural change of the orthorhombic beta-$Ag_2Se$ phase with temperature, differential scanning calorimetry (DSC) and high-temperature XRD studies were performed (cf. Fig. 2a–c). Both measurements reveal that the room temperature (RT) orthorhombic beta-$Ag_2Se$ phase undergoes a structural transition to the cubic alpha-$Ag_2Se$ phase with space group $Im\bar{3}m$ at approximately $T = 410$ K. An endothermic peak at $T = 410$ K during heating and an exothermic peak at $T = 368$ K during cooling are observed in the DSC cycle, which indicates these reversible phase transitions of $Ag_2Se$. The Rietveld refinements of the XRD patterns collected at RT and at 433 K further confirm the phase transition and provide insight into the crystal structures (cf. Fig. 2c). In the beta-$Ag_2Se$ lattice, both Ag and Se atoms occupy Wyckoff position 4a ($x,y,z$), whereas in alpha-$Ag_2Se$ they are located in two inequivalent positions 12 d (1/4, 1/2, 0) and 2a (0, 0, 0), respectively. The lattice parameters of the RT beta-$Ag_2Se$ lattice are plotted for different 'S' contents in Fig. 2d–g. A slight lattice expansion is observed for $x = 0.02$, followed by contraction for $x > 0.02$. This behavior is attributed to a possible suppression of Ag-vacancy formation and subsequent migration of excess Ag into interstitial sites. However, at high doping levels ($x > 0.02$), the substitutional effect of smaller $S^{2-}$ ions dominates, resulting in lattice contraction. Consistently, this structural evolution

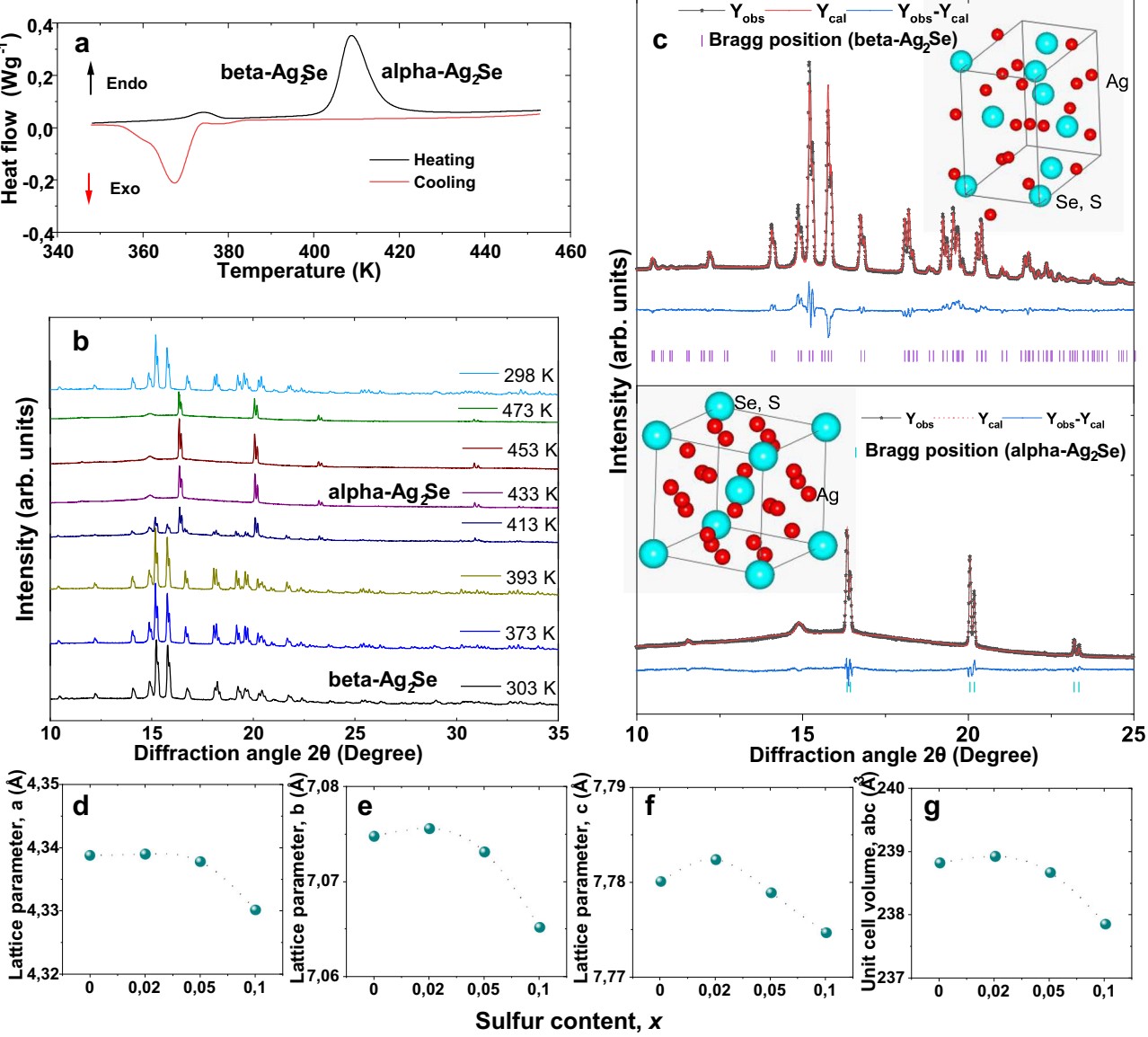

**Fig. 2 | Thermal and crystallographic analysis of printed $Ag_2(Se_{1-x}S_x)_{1.05}$ films.** **a** DSC heating and cooling cycle of the printed film with $x = 0.02$. **b** Temperature-dependent XRD patterns of the printed $Ag_2(Se_{0.98}S_{0.02})_{1.05}$ film. **c** Rietveld refinement of XRD patterns of the beta-$Ag_2Se$ (at 303 K) and alpha-$Ag_2Se$ phase (at 433 K). **d–g** The variation of lattice parameters with sulfur substitutions.

correlates with the trends in the transport properties of the S-doped $Ag_2Se$ printed films.

## Chemical, microstructural and elemental analyses of printed $Ag_2(Se_{1-x}S_x)_{1.05}$ films

Firstly, the survey XP spectra was measured for $x=0.02$ printed film, and peaks corresponding to oxygen ($O1s$), carbon ($C1s$), silver ($Ag3d$) and selenium ($Se3p/Se3d$) could be recognized. Since the amount of sulfur is small, corresponding peaks are not recognized in the survey XP spectra. Then, the detailed $Ag3d$, $Se3d$, and $Se3p/S2p$ XP spectra were recorded. In the latest case, the measurements were carried out using synchrotron-based XPS at the HESGM beamline of the synchrotron facility BESSY II (Helmholtz-Zentrum Berlin, Berlin). The detailed $Ag3d$ spectrum (cf. Fig. 3a) consists of one doublet with a fixed splitting of 6 eV, keeping the intensity ratio of 3:2 between Ag $3d_{5/2}$ and Ag $3d_{3/2}$ lines. The position of Ag $3d_{3/2}$ lines at 367.9 eV is in good agreement with the $Ag^+$ oxidation state, which is typical for $Ag_2Se$[8,15,19]. A very small additional component (corresponding to the $Ag^0$ state) at 368.5 eV has also been observed in the measured spectrum. Since its intensity is on

the level of experimental errors, the variation of the area parameters in the range between 2 and 10 % during the fitting procedure does not result in any visible changes in the envelope spectra. The detailed $Se3d$ spectrum (cf. Fig. 3b) was deconvoluted with two $Se3d$ doublets, applying the splitting of 0.86 eV, intensity ratio of 3:2 in both cases. The binding energies of the Se $3d_{5/2}$ components at 53.4 eV and 55.9 eV are in good agreement with previous experiments, corresponding to $Se^{2-}$ and $Se^0$ oxidation states[8,15]. From the intensity comparison of both components (~12:1) the contribution of $Se^0$ of 7 % is estimated. The detailed $Se3p/S2p$ spectrum is presented in Supplementary Fig. S3a (open symbols). The shape of this spectrum is rather complicate, but its analysis could be partially simplified using $Se^{2-}/Se^0$ parameters obtained from $Se3d$ spectrum (cf. Fig. 3b). The corresponding $Se3p$ components are presented with green ($Se^{2-}$) and blue ($Se^0$) curves. The rest of the intensity could be fitted with three $S2p$ doublets corresponding to $S^{2-}/S^0/S^{\delta+}$ states, which are presented in light blue, magenta and orange curves. The presence of the $S^{2-}$ component results from a partial substitution of selenium atoms with sulfur. The $S2p^{2-}/Se3p^{2-}$ intensity ratio is about 5%, which is in good agreement with the

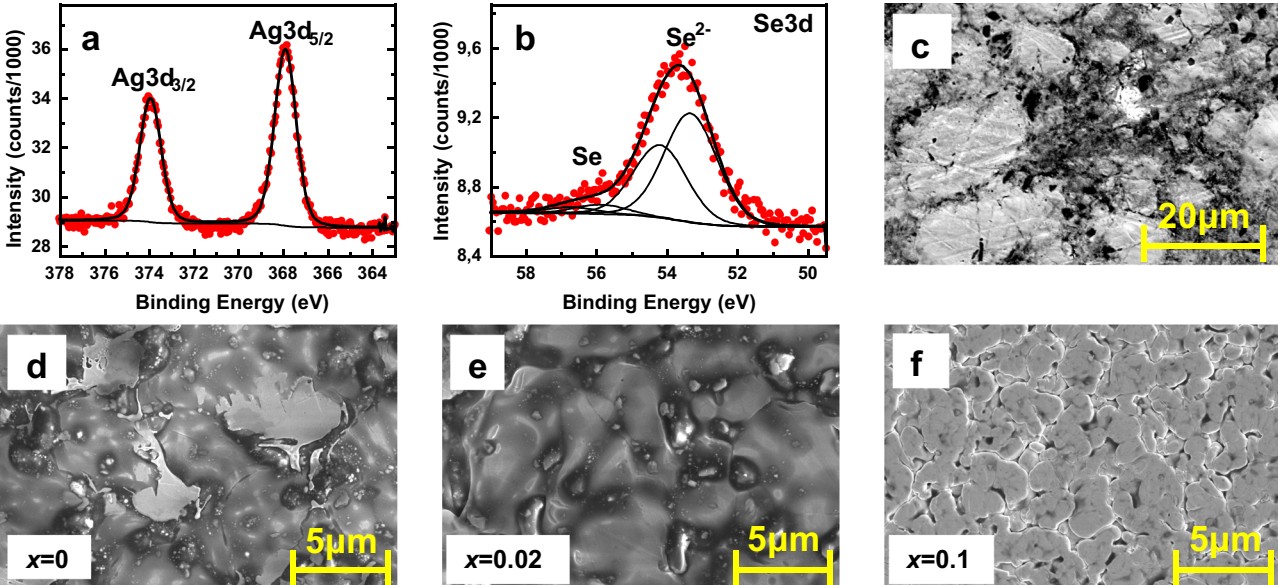

**Fig. 3 | XPS and SEM analysis of printed $Ag_2(Se_{1-x}S_x)_{1.05}$ films. a** Ag3d and (**b**) Se3d. Red circles - experimental spectra, thick black line – envelope spectra. The thin black lines in (**b**) correspond to individual doublets of $Se^0/Se^{2-}$ components. **c** SEM image of the sintered and non-pressed $Ag_2Se$ film. SEM images of the sintered and pressed $Ag_2(Se_{1-x}S_x)_{1.05}$ printed film. **d** $x = 0$. **e** $x = 0.02$. **f** $x = 0.1$.

expected degree of sulfur substitution. The presence of two other components (neutral $S^0$ and oxidized $S^{\delta+}$) was unexpected and could be explained with a co-existence of an unreacted sulfur layer on the $Ag_2(Se_{1-x}S_x)_{1.05}$ film. In order to check this supposition more surface-sensitive XPS measurements were repeated with an excitation energy of photons of 300 eV. For these conditions, the signal was collected from a thickness of 1–2 nm only. The corresponding $Se3p/S2p$ spectrum is shown at inset in Supplementary Fig. S3a. The intensity of "bulk" $Se3p_{3/2}$ component is drastically decreased, but $S2p^0/S2p^{\delta+}$ peaks are still clearly visible, confirming our supposition. Figure 3c shows the non-sintered undoped printed film. Figure 3d–f present the morphology and microstructure of the printed $Ag_2(Se_{1-x}S_x)_{1.05}$ films ($x = 0$, 0.02, and 0.1, respectively). The scanning electron microscope (SEM) images suggest that a relatively smooth surface is formed due to the hot pressing. At high magnification, some micro-pores become visible, which are probably created due to the expulsion of organic ingredients from the surface of the samples during sintering. Supplementary Fig. S2 illustrates the distribution of the chemical composition of the $Ag_2(Se_{0.98}S_{0.02})_{1.05}$ film. The elemental maps of Ag and Se reveal a relatively homogeneous distribution (cf. Supplementary Fig. S2). To quantify the amount of 'S' in the printed $Ag_2(Se_{0.98}S_{0.02})_{1.05}$ film, carrier gas hot extraction (CGHE) analysis was performed. The compound was melted under a carrier gas, and the existing 'S' reacted to form $SO_2$. The quantitative determination of three samples was then done using selective infrared detection. The 'S' content was measured to be 0.47 at. % with a limit of quantification of 0.0143 at %, a standard deviation of 0.0030, and a measurement uncertainty of ± 0.0383. The measured 'S' content is slightly lower than the nominal value, which could be attributed to the measurement limitations of the CGHE method for inorganic chalcogenide film. Here, the analyzer was calibrated using organic 'S' standards, which could underestimate the quantification. In addition, a noticeable carbon signal was also detected in the sample due to decomposition of organic ingredients. This may have affected the measurement by altering the combustion atmosphere. Furthermore, the elemental composition of the printed $Ag_2(Se_{0.98}S_{0.02})_{1.05}$ film was investigated by wavelength-dispersive X-ray spectroscopy (WDX) using electron probe microanalysis (EPMA) at four different probe points (cf. Supplementary Fig. S3b and Supplementary Table S1). The quantitative results, reported in weight percent (wt. %)

**Table 1 | EPMA-WDX analysis of Ag, Se, and S in printed $Ag_2(Se_{0.98}S_{0.02})_{1.05}$ film**

| Element | wt. % (mean ± SD) | at. % (mean ± SD) |
|---|---|---|
| Ag | 68.12 ± 3.29 | 66.64 ± 1.19 |
| Se | 24.51 ± 2.12 | 32.70 ± 1.22 |
| S | 0.200 ± 0.022 | 0.65 ± 0.10 |
| Total | 92.82 ± 5.11 | 100.00 ± 0.00 |

and atomic percent (at. %), confirm a consistent distribution of Ag, Se, and S across the probe points. The Ag amount varies between 65.01 and 67.86 at. %, Se between 31.45 and 34.37 at. %, and S between 0.53 and 0.76 at%. The presence of sulfur indicates a reasonably uniform distribution at the micrometer scale. The small compositional variations are probably due to film porosity and matrix effects in the printed film. The nitrogen-free (NF) value of 0.03 at all probe points indicates minor background contributions. In addition, a negligible trace of Se-rich phase is observed in the printed film (light gray region in Supplementary Fig. S3b). To provide a statistical overview of all elemental compositions, the mean values and standard deviations (SD) from the four probe points are calculated and presented in Table 1. The presence of Ag, Se and S in the printed film are in good agreement with the nominal composition of $Ag_2(Se_{0.98}S_{0.02})_{1.05}$, confirming an effective sulfur incorporation within the printed material. The total measured Ag, Se and S content accounts to 92.82 ± 5.11 wt. %, which is slightly below 100 wt. %. This deviation is attributed to the presence of porosity and residual organic ingredients in the printed film, which are not detected by EPMA-WDX.

## Mechanical properties of printed $Ag_2(Se_{1-x}S_x)_{1.05}$ films

To study the robustness of the printed films, the mechanical properties of three printed samples with $x = 0$, 0.02, 0.05, and 0.1 were studied by measuring their resistance with bending cycles for different bending diameters, as shown in Fig. 4. Neither did the samples show any visible crack or deformation after the bending test, nor was peeled off from the substrate. The initial resistances of the samples were different because of the different doping levels and sample thicknesses. Figure 4b–d shows the normalized-resistance vs bending cycle plots of

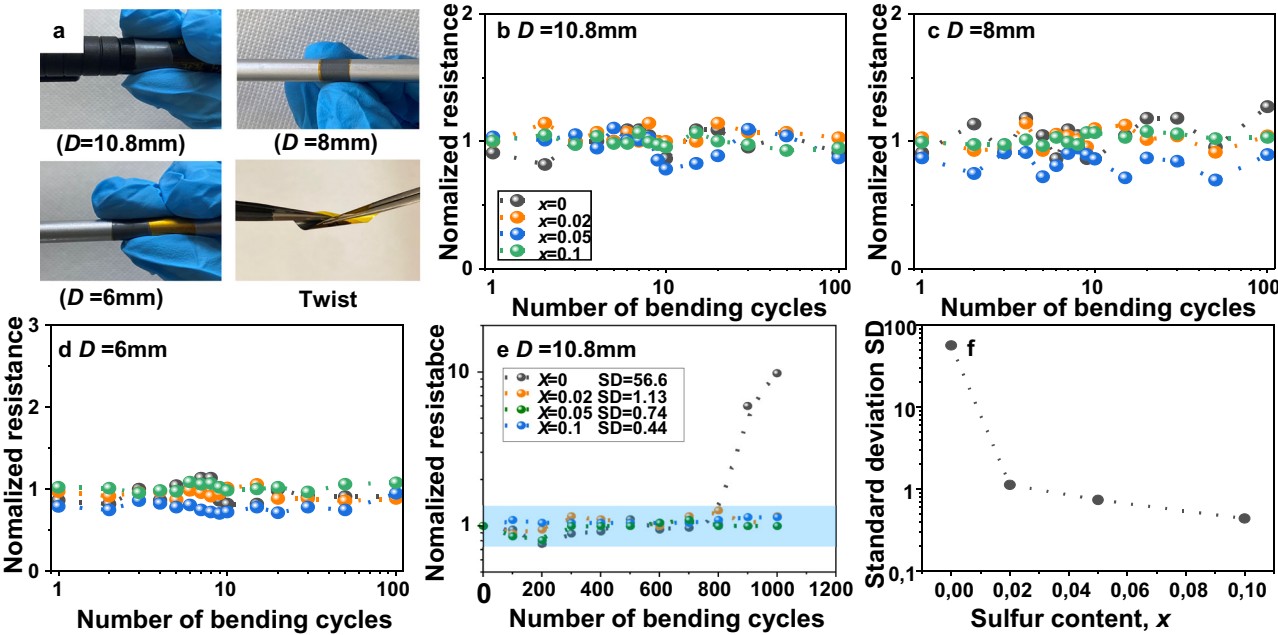

**Fig. 4 | Mechanical flexibility test of n-type Ag₂(Se₁₋ₓSₓ)₁.₀₅ films. a** Images of printed $Ag_2(Se_{0.98}S_{0.02})_{1.05}$ film during bending and twisting conditions. Normalized resistance and bending cycle curve with different bending diameters, $D$; (**b**) 10.8 mm, (**c**) 8 mm, (**d**) 6 mm. **e** Normalized resistance of the printed films for $D = 10.8$ mm during 1000 bending cycles. **f** Variation of standard deviation in normalized resistance data with '$x$' during 1000 bending cycles.

the hot-pressed films for three different bending diameters, $D = 10.8$ mm, 8 mm, and 6 mm. The resistance increases by less than 30% after 100 bending cycles for all samples. These results underline that hot pressing after printing improves the flexibility and adhesion of the samples. Less thickness and higher density due to compaction contribute to the improvement of flexibility. In principle, sulfur doping could improve the flexibility of the $Ag_2Se_{1.05}$ film. However, as the normalized resistance varies within a similar range, the mechanical behavior is similar for all samples. This could be because we are still far away from the limitation of bending cycles; all the samples could tolerate many more cycles than 100. In addition, to check the mechanical properties, we also checked the change in resistance after twisting the films (cf. Supplementary Fig. S4). After twisting, the resistance of the samples remains almost the same. Furthermore, bending tests were carried out for 1000 cycles on all films at a $D = 10.8$ mm to assess their mechanical tolerance (cf. Fig. 4e, f). The normalized resistance of the undoped $Ag_2Se$ film increases abruptly after 800 bending cycles due to crack formation, whereas the printed films with $x > 0$ maintain stable resistance up to 1000 bending cycles (cf. Fig. 4e). To quantify the effect of 'S' substitution on mechanical flexibility, the standard deviation (SD) of normalized resistance is plotted as a function of '$x$' (cf. Fig. 4f). The SD decrease with increasing '$x$', indicating improved ductility. The SD values vary between 1 to 0.40 for $x > 0.02$ films. These results confirm the excellent durability of the S-doped printed films under repeated bending.

## TE performance of printed Ag₂(Se₁₋ₓSₓ)₁.₀₅ films

The temperature-dependent TE performance of the printed $Ag_2(Se_{1-x}S_x)_{1.05}$ films for the different compositions was studied over 303–463 K, as shown in Fig. 5. The negative values of the Seebeck coefficient $\alpha$ and the Hall coefficient $R_H$ for all samples are due to the electrons being the majority carriers. $\alpha$ decreases with increasing sulfur content (cf. Fig. 5a), which is due to the increase in carrier density $n_H$ as confirmed by the Hall measurement (cf. Fig. 5d). The highest $\alpha$ of ~145–150 μV K⁻¹ is achieved at room temperature for the films with low sulfur concentrations $x \leq 0.02$. A sharp drop in $\alpha$ is observed at a temperature $T \approx 410$ K for all samples. This is attributed to the phase

transition of $Ag_2Se$ from an orthorhombic (beta-phase) to cubic (alpha-phase), consistent with DSC and XRD results. Figure 5b shows the temperature-dependent electrical conductivity $\sigma$. In conjunction with the temperature-dependent carrier density, a semiconducting behavior of all the samples is confirmed. $\sigma$ increases with increasing the sulfur content up to $x = 0.02$ due to an increase in $n_H$ (cf. Fig. 5c). It decreases for $x > 0.02$, due to the significant reduction in carrier mobility $\mu_H$ (cf. Fig. 5e). A maximum conductivity of 650 S cm⁻¹ at 300 K was observed for the printed film with $x = 0.02$. Similar to $\alpha$, $\sigma$ also abruptly declines due to the phase transition at $T \approx 410$ K. The substitution of Se by S is expected to alter the defect chemistry of the $Ag_2Se$ lattice. At low-level doping ($x = 0.02$), S incorporation is assumed to suppress the formation of Ag vacancies and facilitates donor-type defects through the accumulation of excess Ag on interstitial sites[20]. As a result, the $n_H$ increases while preserving high mobility $n_H$ (cf. Fig. 5c, e). Although the ionic radius of $S^{2-}$ is smaller compared to $Se^{2-}$, lattice distortion at $x = 0.02$ is not significant enough to induce notable electron scattering. Consistently, Rietveld refinement shows a small increase in the lattice volume for $x = 0.02$, followed by a decrease at $x > 0.02$. Despite the smaller radius of $S^{2-}$, this slight expansion can be attributed to the presence of excess Ag on interstitial sites. However, the substitutional ionic-radius mismatch starts to dominate for $x > 0.02$, resulting in lattice contraction and higher carrier scattering. Despite lattice contraction due to the dominant substitutional effect, the continuous rise $n_H$ for $x > 0.02$ can be attributed to the further evolution of donor-type defects. Consequently, the carrier mobility $\mu_H$ decreases while retaining higher $n_H$. The maximum carrier mobility of 1050 cm² V⁻¹ s⁻¹ is observed at room temperature for the film with $x = 0.02$. The mobility decreases with temperature in agreement with an enhanced electron-phonon scattering[21]. Hence, the non-monotonic behavior in both the electronic transport properties and lattice parameters arises from the interplay between the possible vacancy suppression, lattice expansion and subsequent lattice distortion, facilitating an optimum TE power factor $PF(\alpha^2\sigma)$ at $x = 0.02$. It is plotted as a function of temperature as shown in Fig. 5f. The maximum power factor of ~16 μW cm⁻¹ K⁻² is obtained for the film with $x = 0.02$ at 360 K, indicating that the inclusion of a small

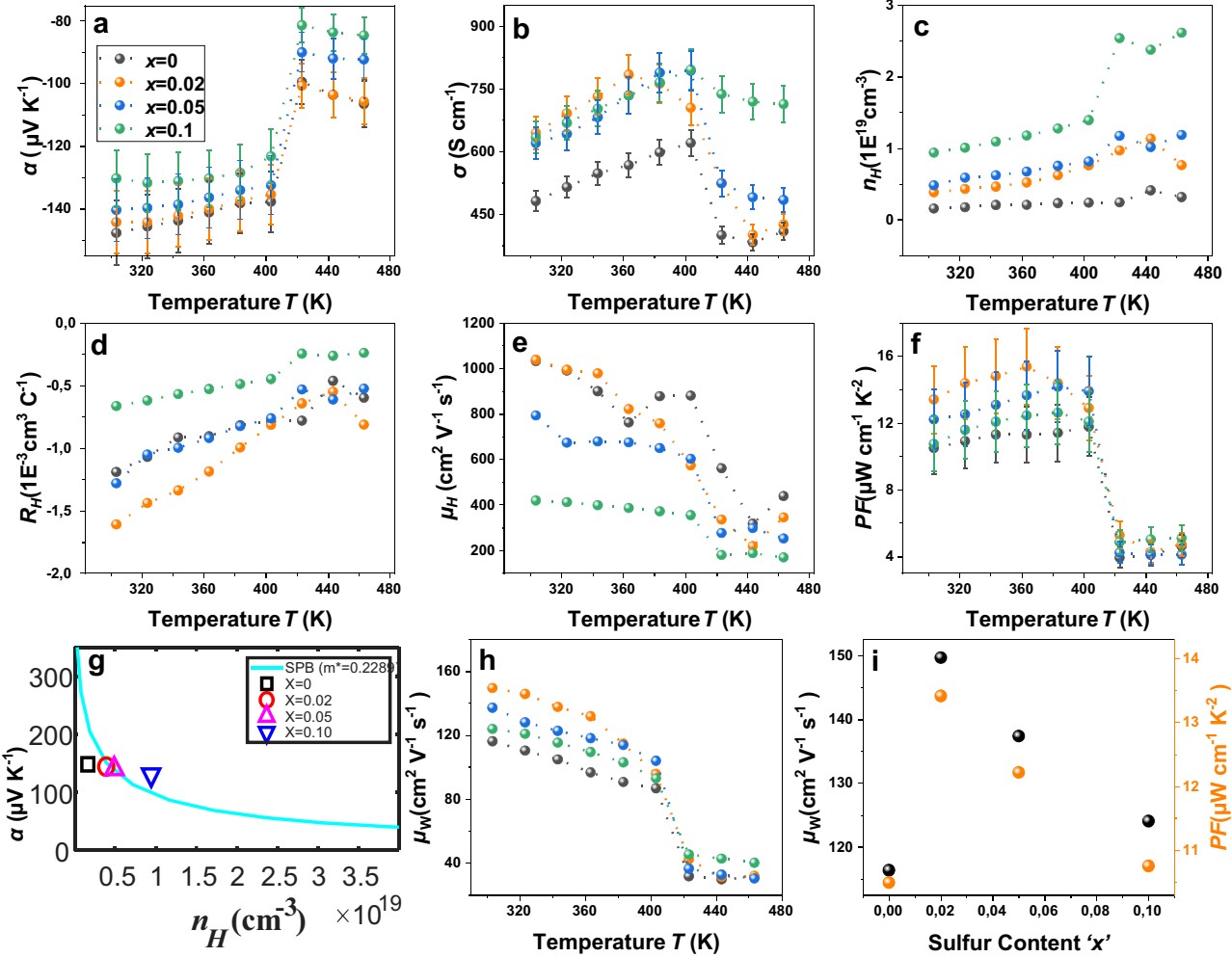

**Fig. 5 | Temperature-dependent TE transport properties of n-type Ag₂(Se₁₋ₓSₓ)₁.₀₅ films. a** Seebeck coefficient $\alpha$, (**b**) Electrical conductivity $\sigma$, (**c**) Charge carrier concentration $n_H$; (**d**) Hall coefficient $R_H$, (**e**) Hall mobility $\mu_H$, (**f**) Power factor $PF$, (**g**) Ioffe-Pisarenko plot for the effective mass of 0.2289 electron mass with experimental data, (**h**) The weighted mobility $\mu_W$, and **i** Variation of room temperature power factor and weighted mobility with S substitution.

fraction of sulfur facilitates the optimum TE performance for Ag₂(Se₁₋ₓSₓ)₁.₀₅. The power factor is enhanced by 40 % for $x = 0.02$ compared to the pristine film. Further, at higher temperatures, the power factor also reflects the phase transition. Thermal conductivity is estimated to be 0.59 Wm⁻¹ K⁻¹ for the $x = 0.02$ film based on the measured value of a non-pressed film. The predicted $ZT$ value of the film for $x = 0.02$ is 0.7 at RT (see Supplementary information Note S4). To gain deeper insight into the transport mechanism, weighted mobility ($\mu_W$) and a theoretical Ioffe-Pisarenko curve have been evaluated using the single-parabolic band (SPB) model. The temperature-dependent weighted mobility $\mu_W$ was calculated from the electrical conductivity and Seebeck coefficient using the following equation[21];

$$\mu_w = \frac{3h^3\sigma}{8\pi e(2m_e k_B T)^{3/2}} \left[ \frac{\exp\left[\frac{|\alpha|}{k_B/e} - 2\right]}{1 + \exp\left[-5\left(\frac{|\alpha|}{k_B/e} - 1\right)\right]} + \frac{\frac{3}{\pi^2}\frac{|\alpha|}{k_B/e}}{1 + \exp\left[5\left(\frac{|\alpha|}{k_B/e} - 1\right)\right]} \right] \quad (1)$$

In addition, the theoretical Ioffe-Pisarenko curve has been calculated by the SPB model with acoustic phonon scattering for an effective mass of $0.2289 m_e$ using the following relation[22,23]

$$\alpha = \frac{k_B}{e} \left( \frac{2F_1(\eta)}{F_0(\eta)} - \eta \right) \quad (2)$$

where the Fermi integral of order $j$ is defined as:

$$F_j(\eta) = \int_0^\infty \frac{\epsilon^j d\epsilon}{1 + \exp(\epsilon - \eta)} \quad (3)$$

The chemical carrier concentration is expressed as:

$$n = 4\pi \left( \frac{2m^* k_B T}{h^2} \right)^{\frac{3}{2}} F_{\frac{1}{2}}(\eta) \quad (4)$$

while the Hall carrier concentration is given by:

$$n_H = \frac{n}{r_H} \quad (5)$$

with the Hall factor

$$r_H = \frac{3}{2} F_{1/2}(\eta) \frac{F_{-1/2}(\eta)}{2F_0^2(\eta)} \quad (6)$$

In these expressions, $\eta$ is the reduced chemical potential, $\epsilon$ is the reduced energy, $k_B$ is the Boltzmann constant, $e$ is the bare electronic charge, $m_e^*$ is the density-of-states effective mass, $T$ is the temperature, and $h$ is the Planck constant. Similar trend is also observed in the temperature dependent $\mu_W$ for all films

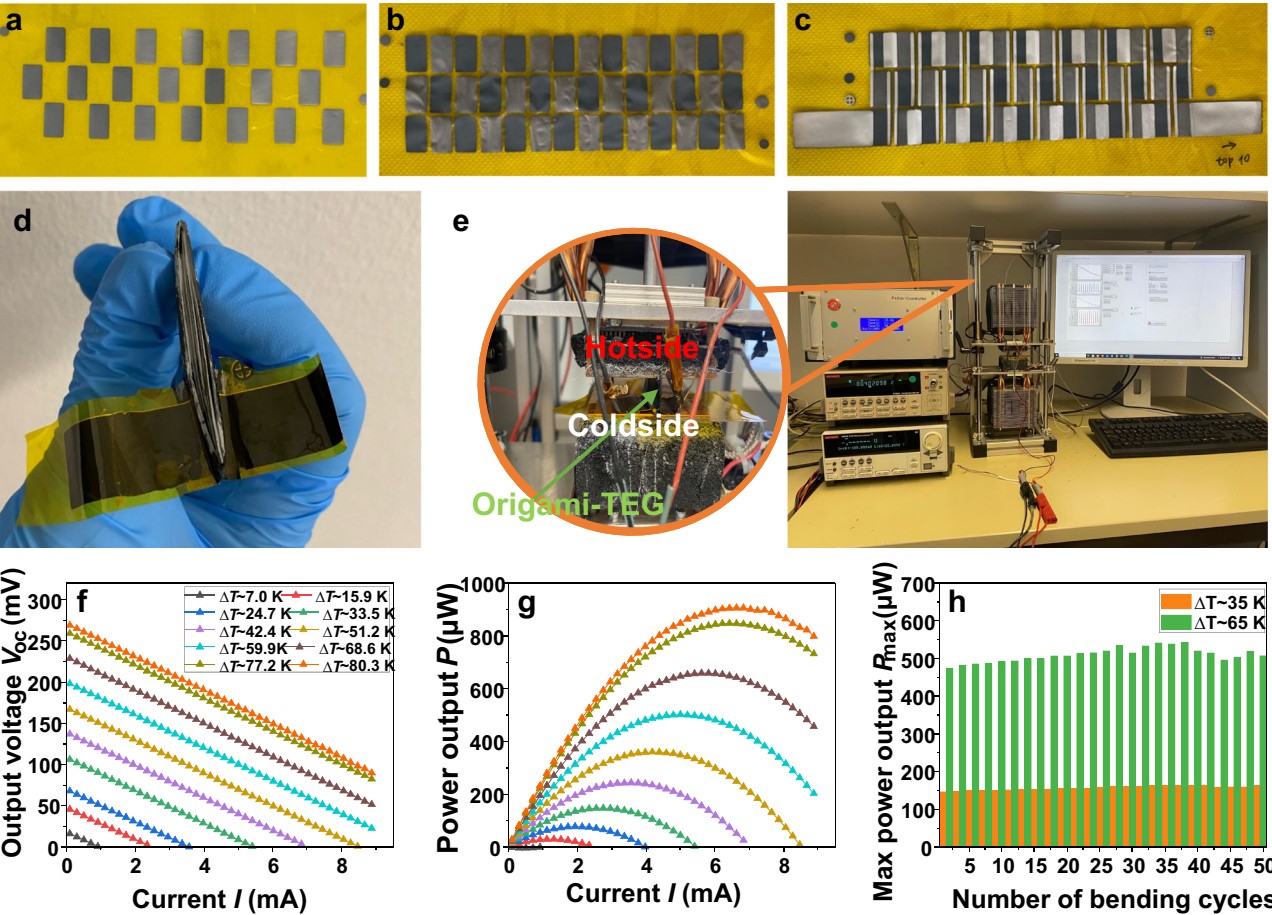

**Fig. 6 | Setup and performance of printed origami-TEG. a** Top view of printed n-type legs. **b** printed p-type legs. **c** both printed legs with electrodes. **d** The origami-TEG after encapsulation and folding. **e** Experimental setup for V-I characterization using a source measuring unit (SMU). **f** V-I characteristics of the origami-TEG for various temperature differences. **g** Temperature-dependent P-I characteristics of the printed origami-TEG. **h** Long-term performance of the origami-TEG upon cycling between two temperature differences. The hot side temperature was cycled between 313 K and 343 K, while the cold side was kept at 278 K.

(cf. Fig. 5h). The $\mu_W$ is one of the key descriptors of electronic transport properties, comprises of $\mu_H$ and effective mass ($m^*$). It is found to be the highest at $x = 0.02$ for the beta-Ag$_2$Se phase, indicating the optimum trade-off between $\mu_H$ and $n_H$. The $\mu_W$ decreases rapidly due to carrier scattering for $x > 0.02$, despite increasing of $n_H$. To further understand the defect chemistry of the Ag$_2$Se lattice upon S substitution, the theoretical Ioffe-Pisarenko curve is shown in Fig. 5g. It shows that the experimental effective masses in all samples lie around the simulated curve, with little deviation found upon high S substitution due to the point defects. Figure 5i shows the room temperature $x$-dependent power factor and $\mu_W$, and confirm that $x = 0.02$ is the optimum S substitution for power generation for low temperature TE devices. To check the reproducibility, two printed Ag$_2$(Se$_{0.98}$S$_{0.02}$)$_{1.05}$ films were prepared from two different batches of inks. The TE performance of films is calculated and compared in Supplementary Fig. S6. It is found that the overall power factor value remains within 10 %. The TE performance of the printed Ag$_2$(Se$_{0.98}$S$_{0.02}$)$_{1.05}$ film was also measured in 5 heating/cooling cycles to evaluate the thermal stability (cf. Supplementary Fig. S6). There is a small hysteresis found between heating and cooling. However, the results show the film is thermally stable and produces similar performance. The TE properties of the printed p-type Bi$_{0.5}$Sb$_{1.5}$Te$_3$ film were also studied and agreed with previous studies. The data is included in the supplementary information (cf. Supplementary Fig. S8).

## Performance of printed origami-TEG

The origami-TEG was fabricated using the developed printed TE material. The detailed fabrication processes and characterization techniques are described in 'Method'. The images of a fully printed origami-TEG are shown in unfolded and folded conditions (cf. Fig. 6c, d). The TEG characterization setup with the origami TEG connected to an SMU, is shown in Fig. 6e. When a temperature difference is maintained at the two sides of the origami-TEG, charge carriers in the n-type and p-type legs move from the hot side to the cold side, generating a voltage. The output voltages were recorded while the current through the device was varied to study its V-I characteristics. The performance of the origami-TEG was studied by sweeping the current from −9 mA to 9 mA with a step width of 0.1 mA (cf. Fig. 6f, g). The maximum $\Delta T$ of 80 K was applied between the hot and cold sides of the TEG. The open circuit voltage ($V_{OC}$) and the power output $P$ of the device increase with increasing $\Delta T$. The fully printed origami-TEG generates a $V_{OC}$ of 269 mV at $\Delta T = 80$ K with a corresponding maximum power output $P_{max}$ of 907 µW. To estimate the maximum power density $p_d$, the cross-sectional area of the origami-TEG is calculated to be 43.2 mm$^2$ by multiplying the device width of 0.911 mm and the length of 47.4 mm. A maximum $p_d$ of 21 W m$^{-2}$ is achieved at $\Delta T = 80$ K. For long-term energy harvesting applications, it is crucial to test the stability of the TEG over an extended period of cycling. Stability tests have been conducted on origami-TEG by running it for 50 consecutive cycles at two different $\Delta T$ of 65 K and 35 K, respectively, as shown in Fig. 6h. Each cycle consists of five current sweeps. A stable output power of ~158.17 µW was

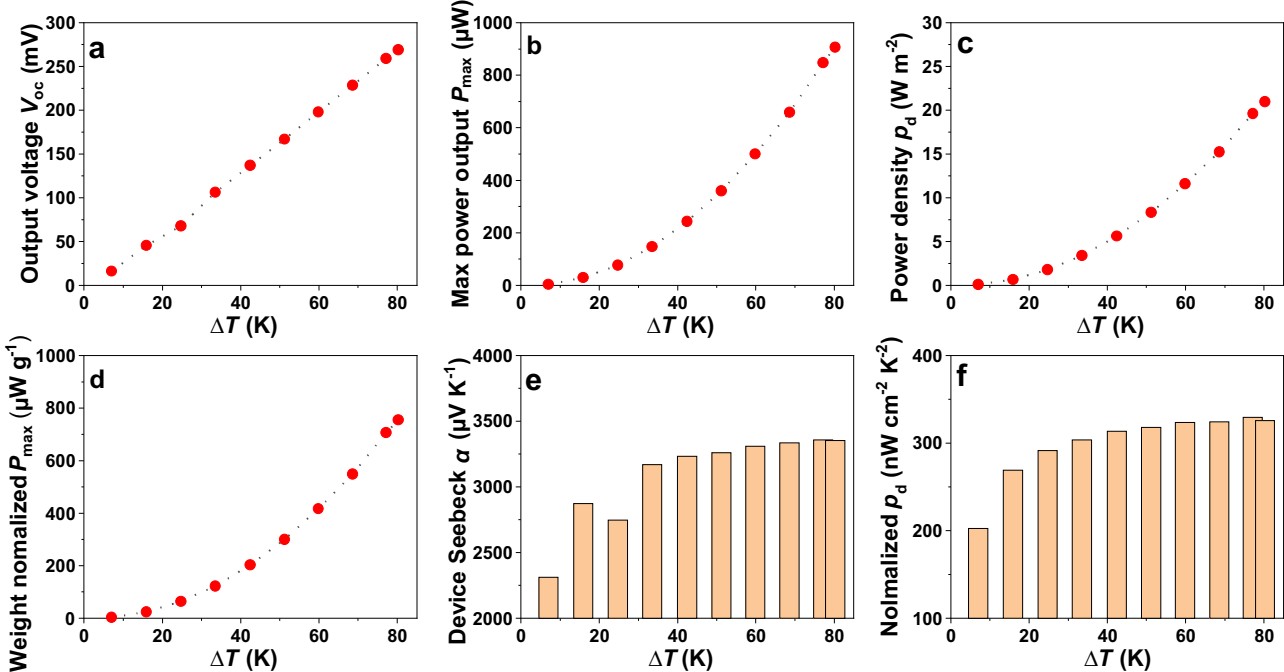

**Fig. 7 | Performance of origami-TEG for different $\Delta Ts$. a** Open circuit voltage $V_{OC}$. **b** Maximum power output $P_{max}$. **c** Power density $p_d$. **d** Weight normalized $P_{max}$. **e** Device Seebeck. **f** Normalized $p_d$.

measured at a $\Delta T$ of ~ 35 K, and ~ 513.14 µW at a $\Delta T$ of ~ 65 K. High stability is indicated by the resulting power output of a fully printed origami TEG, with a relative standard deviation of ~ 3 %. The $V_{OC}$, $P_{max}$ and $p_d$ of the printed origami-TEG increase with increasing $\Delta T$ as shown in Fig. 7a–c. From these data, a high weight-normalized power density of 800 µW g$^{-1}$ can be derived for $\Delta T = 80$ K (cf. Fig. 7d). This is a high value for a fully printed TEG, highlighting its strong potential in energy harvesting for IoT and wearable applications. Although the $\alpha$ for n-type TE material decreases with increasing temperature, the device's Seebeck coefficient remains almost constant at higher $\Delta Ts$ after increasing initially, as shown in Fig. 7e. This is due to an increase in $\alpha$ with increasing temperature for the Bi$_{0.5}$Sb$_{1.5}$Te$_3$ material used for the p-type leg. This trend of thermo-voltage in a printed-TEG is important to maintain high performance at elevated temperatures. The origami-TEG exhibit similar normalized $p_d$ at a $\Delta T > 20$ K as shown in Fig. 7f.

**Simulated results of printed origami-TEG**

The fully printed origami-TEG was modeled and simulated to compare its theoretical performance with experimental results. Two-folded strips, consisting of three thermocouples, were modeled in COMSOL Multiphysics 6.2. For better visualization, the view scale in the Z-direction is magnified by a factor of 50 (cf. Supplementary Fig. S10). COMSOL Multiphysics was selected for this analysis due to its capability to handle multiple coupled physics phenomena. Within the software, respective material properties were assigned to all the components of the origami-TEG: Kapton sheet as the substrate, n-type Ag$_2$(Se$_{0.98}$S$_{0.02}$)$_{1.05}$ film, and p-type Bi$_{0.5}$Sb$_{1.5}$Te$_3$ as the TE legs, carbon paste as an interface layer between the electrodes and TE legs, silver as the electrode, and carbon tape to encapsulate the whole TEG device. Structural parameters of the origami-TEG device used in COMSOL are given in Supplementary Table S2. The simulation was conducted for several $\Delta Ts$ across the origami TEG as observed in the experiments. In the heat transfer module, an additional thermal resistance of 0.002 K W$^{-1}$ was applied at the interfaces between the copper block and the device surface. After defining the thermal boundary conditions, electrical boundary conditions were

applied to the respective terminals – one as the active terminal and the other as the ground. In Multiphysics, Joule heating was enabled for all the domains carrying current, and the thermoelectric effect was only considered for n-type and p-type domains. The current value was varied from 0 to 10 mA to get the maximum power point of the TEG model. We used a structured mesh type with an extremely fine element size to ensure the accuracy of the simulations and used the fully coupled solver. Figure 8 illustrates the simulation results, including the 3D distributions of electric potential and temperature (Fig. 8a, b), the simulated and experimental V-I and P-I characteristic curves of the TEG (Fig. 8c, d), as well as the simulated and experimental $V_{OC}$ and $P_{max}$ as functions of $\Delta T$ (Fig. 8e, f). From the simulated V-I characteristics, an open circuit voltage $V_{OC}$ of 275 mV is observed at $\Delta T = 80.3$ K. This agrees well with the experimentally observed value of 270 mV. The minor differences between the simulation and experimental results can be due to a higher thermal resistance caused by imperfect mechanical contact and the presence of an air gap between the TEG and the measurement setup. In addition, the radiation thermal losses and imperfections of the real device due to printing roughness, porosity, and fabrication errors were not considered in the COMSOL simulation.

In summary, this study reports a fully printed origami TEG as a promising alternative power source for low-power electronics. We carried out an extensive investigation encompassing TE material development and device fabrication. We developed a high-performance Ag$_2$(Se$_{1-x}$S$_x$)$_{1.05}$-based n-type flexible TE film via engineering non-stoichiometric defects and sulfur inclusion. Using this material, an ultra-thin (< 1 mm) fully printed origami TEG was fabricated. The printed TEG achieved a maximum power output of 906 µW at $\Delta T = 80$ K, with a peak power density of 21 W m$^{-2}$, setting a record for fully printed TEGs. These results pave the way for a future maintenance-free power source for IoT applications.

## Methods
### Materials
Silver powder (200 mesh, ≥ 99.9% trace metal basis, Sigma-Aldrich), selenium powder (200 mesh, ≥ 99.5%, Thermo Fisher Scientific), sulfur

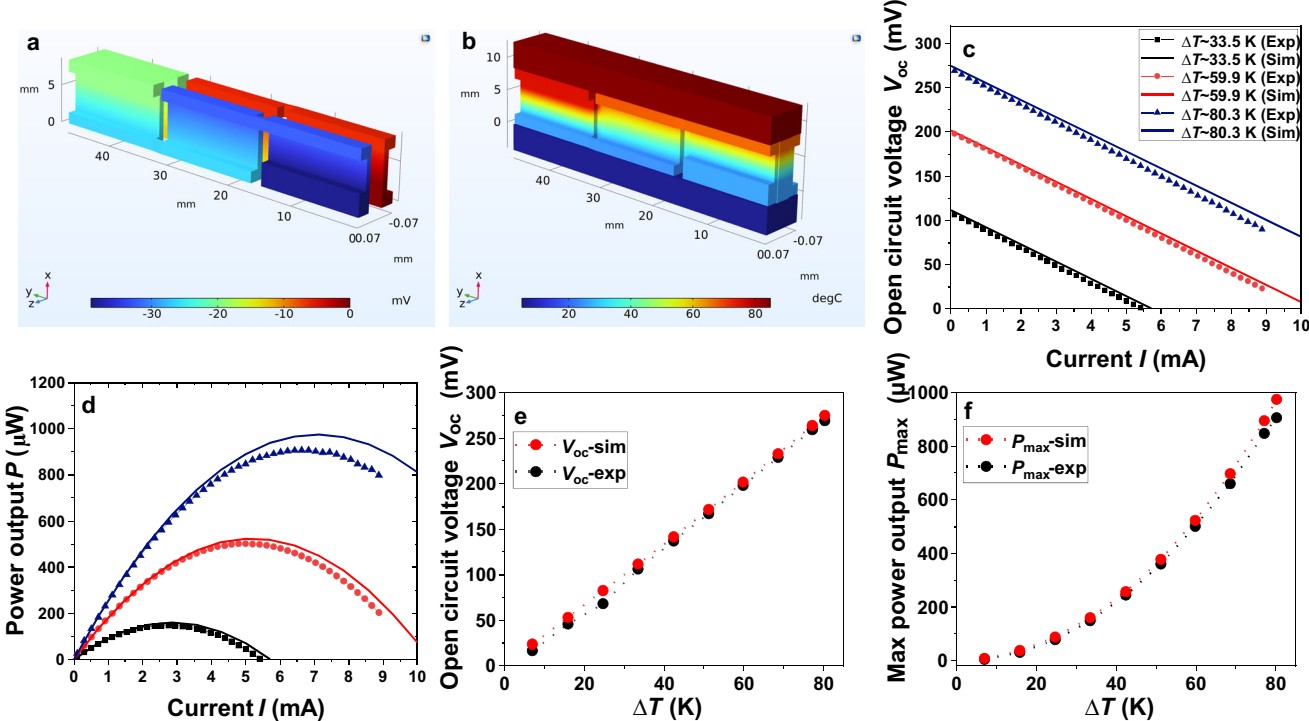

**Fig. 8 | COMSOL simulated model of printed origami-TEG. a** Electric potential map of origami-TEG. **b** Thermal map of origami-TEG. **c**–**f** Comparison of experimental and simulated data for different temperature differences: **c** $V$-$I$ characteristics at $\Delta T$s. **d** $P$-$I$ characteristics. **e** Open circuit voltage $V_{OC}$. **f** Max power output $P_{max}$.

powder (100 mesh, ≥ 99.5%, Alfa Aesar), polyvinylpyrrolidone (PVP) (average Mw ~ 40000, Sigma Aldrich), N, N-Diethylformamide (DEF) (Sigma-Aldrich), terpineol (Sigma Aldrich), Kapton (Dietrich Müller GmbH), Carbon paste (Dycotec DM-CAP-4701S), Silver conductive ink (Thermo Fischer), Heat conductive paste (IBF-FEROTHERM 4).

### Preparation of printable $Ag_2(Se_{1-x}S_x)_{1.05}$ inks and TE films

The preparation method of the $Ag_2(Se_{1-x}S_x)_{1.05}$ inks follows the previously reported work[8]. The printable inks were prepared by wet milling of inorganic particles and an organic solution. For the compositions $Ag_2(Se_{1-x}S_x)_{1.05}$, where $x$ = 0, 0.02, 0.05, and 0.1, elemental Ag, Se, and S powders were weighed in the corresponding ratios and used as inorganic TE particles. The organic solution was prepared by mixing an organic binder, PVP, with the solvent mixture, resulting in an 8 wt.% solution. As a solvent, we use a blend of terpineol and DEF in a 4:1 ratio. The solution mixture was stirred at 1000 rpm for 10 min to dissolve the PVP and achieve a homogeneous solution. Finally, the TE n-type inks were prepared by wet milling the TE particles in the prepared organic solutions using a 120 mL zirconia jar with 10 mm diameter zirconia balls. The weight ratio of TE particles to the organic solution was set at 4:1. The weight ratio of TE particles to milling balls was 3:1. The ball mill jars were sealed and flushed with a steady, gentle stream of nitrogen gas ($N_2$) for 5 min before milling. Finally, the jars were loaded into a Fritsch Planetary Mill (PULVERI-SETTE 5 premium line) and ball-milled at 350 rpm for 60 min. The produced n-type TE inks were stored under inert conditions for subsequent printing. The films of n-type ink were printed on Kapton substrates using screen printing and then sintered at 473 K for 10 min in ambient air. After sintering, the printed films were hot-pressed at 573 K under ~70 MPa. The TE performance of the n-type printed films for all compositions was studied and optimized.

### Fabrication of fully printed origami-TEG

The origami-TEG was fabricated using the optimized printed $Ag_2(Se_{0.98}S_{0.02})_{1.05}$ film for the n-type legs in combination with the

$Bi_{0.5}Sb_{1.5}Te_3$-based film for the p-type legs. The details of the synthesis and fabrication of the p-type legs were reported previously[8]. The TEGs were screen-printed on a flexible Kapton substrate using different layouts for p-type, n-type, interface layer, and electrode. Kapton was selected due to its excellent thermal stability, electrical insulation, mechanical flexibility, and suitability for high-temperature processes and applications. A carbon paste was used as the interface layer, which reduces interfacial resistance. Silver ink was used to fabricate the electrodes. Screen printing was chosen for its ability to produce uniform and precisely patterned films, suitable for large-area deposition and enabling high-quality layers with controlled thickness. This technique involves pressing the ink through a mesh stencil onto the substrate, providing design versatility and high reproducibility. The fabrication process began with screen printing of the p-type ink, which was subsequently sintered on a hotplate in a glovebox at 623 K for 30 min to enhance electrical conductivity and stability. Next, the n-type $Ag_2(Se_{0.98}S_{0.02})_{1.05}$ ink was printed and sintered at 473 K on a hotplate for 10 min in ambient air, to form the phase and achieve optimal TE performance. Subsequently, a carbon layer was screen printed, followed by the printing of silver ink for electrical contacts. After printing, the device was hot-pressed to further enhance electrical and mechanical properties. The printed structure was then encapsulated with carbon tape to improve mechanical stability and insulate the top surface. Finally, the device was folded along predefined lines into an origami-inspired three-dimensional structure, resulting in a height of 8 mm. The routine of the fabrication process of the origami-TEG is shown in Fig. 9. The total width of the device was measured to be 911 μm, closely matching the theoretical value of 917 μm, which was calculated by summing the average thicknesses of all 14 folded layers. This includes the Kapton substrate (24 μm), the p-type layer (17 μm), the n-type layer (14 μm), the carbon layer (12 μm), the silver layer (10 μm), and the encapsulation layer (4 μm). Using the measured width, the device area was calculated to be 43.2 mm² (0.911 mm × 47.4 mm). The origami TEG has a fill factor of 23%, which is defined as

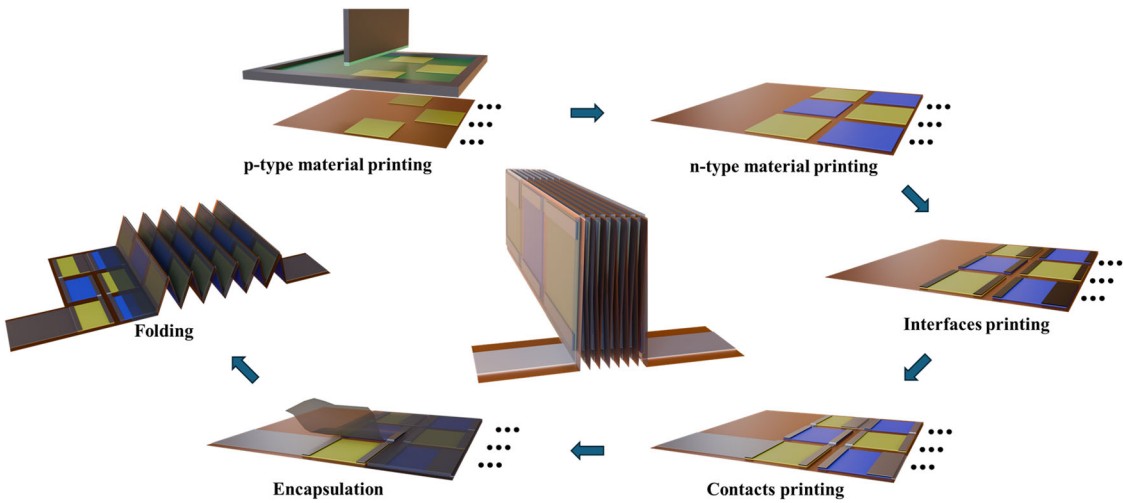

**Fig. 9 | Schematic representation of origami-TEG fabrication steps using screen printing.**

the ratio of the total volume of TE materials to the overall device volume.

## Characterization techniques

The electronic transport properties of p- and n-type printed TE films are measured using a Hall measurement system, Linseis HCS 10. The temperature-dependent transport parameters Seebeck coefficient ($\alpha$), electrical conductivity ($\sigma$), Hall coefficient ($R_H$), and charge carrier concentration ($n_H$) were measured in the range from ambient temperature to 463 K. The relative errors associated with the $\alpha$ and $\sigma$ measurements were 7% and 6%, respectively. The crystallographic structures and phases of the samples were studied by the X-ray diffraction (XRD) technique in Bragg–Brentano geometry using Cu-K$_{\alpha1,2}$ radiation in a Bruker D8 diffractometer with a Lynxeye XE detector. The XPS analyses were done under an ultra-high vacuum with a base pressure of $9 \times 10^{-10}$ mbar, and core-level spectra were recorded under normal emission using Mg-K$_\alpha$ radiation (1253.6 eV) with a Scienta R4000 hemispherical electron analyzer. For a detailed investigations of Se3p/S2p region the measurements were carried out using synchrotron-based XPS at HESGM beamline of synchrotron facility BESSY II, which is a part of Helmholtz-Zentrum Berlin for Materials and Energy (HZB), Berlin. For a quantitative analysis of the experimental spectra, the CasaXPS software was utilized. The line correction was adjusted to the O1s peak at 532 eV, and all spectra were fitted with a Voigt profile (10% of Lorentz contribution) using Shirley background. Microstructural and compositional analyses were performed on n-type printed films using an FEI Quanta 650 environmental scanning electron microscope (SEM) equipped with a Schottky field emitter. Energy dispersive X-ray spectroscopy (EDX) measurements were taken using a Bruker EDX Quantax 400 with a solid-state detector operating at a voltage of 15 kV. To quantify the sulfur content in the printed Ag$_2$(Se$_{0.98}$S$_{0.02}$)$_{1.05}$ film, carrier gas hot extraction (CGHE) analysis and electron probe microanalyses (EPMA) using a JEOL JXA-8200 wavelength-dispersive electron microprobe were performed. EPMA analyses were performed at an accelerating voltage of 15 kV, a beam current of 10 nA, using a focused beam diameter of 1 μm. Peak and background counting times were optimized for low-level detection of chalcophile elements (50 s on-peak, 25 s background). Matrix corrections were applied using the φ(ρz) routine implemented in the JEOL software. Well-characterized standards were used for elemental calibrations, i.e., Galena (PbS) for sulfur as well as synthetic metallic Ag for silver, and synthetic metallic Se for selenium analysis. Detection limits under these analytical conditions were generally below c. 300 ppm for Ag and Se and

below c. 100 ppm for S. Analyses were monitored for potential beam damage and element loss; none was detected under the chosen beam conditions. Repeated measurements of standards and internal reference materials yielded analytical reproducibility better than 1% relative for major elements and 5% relative for minor elements. The mechanical flexibility of the printed films was assessed by measuring the change in resistance with bending cycles for different bending radii. The performance of the origami-TEG was studied using the maximum power point tracking method by a Keithley Source Measuring Unit 2601B sweeping a current from −10 mA to +10 mA. The origami-TEG was characterized by maintaining a temperature difference ($\Delta T$) from 5 K to 80 K between the top and bottom sides.

## Data availability

The authors declare that the data supporting this study are available within the paper and its supplementary information files. The source data that support the findings of this study have been deposited in the Zenodo database (https://doi.org/10.5281/zenodo.18174258).

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

## Acknowledgements

The authors acknowledge funding by the European Research Council, grant 101097876 – ORTHOGONAL to N.L, Z.W, A.K.V, M.I.K, L.F., U.L, and M.M. The authors wish to acknowledge the Deutsche Forschungsgemeinschaft (DFG, German Research Foundation) under Germany´s Excellence Strategy via the Excellence Cluster 3D Matter Made to Order (EXC-2082/2) to Z.W., M.I.K., Y.M.E., U.L., and M.M. This project was also funded by the Federal Ministry of Education and Research (BMBF) and the Baden-Württemberg Ministry of Science as part of the Excellence Strategy of the German Federal and State Governments to M.M. The German Federal Environmental Foundation (Deutsche Bundesstiftung Umwelt - DBU), through the DBU Ph.D. scholarship program, also supported this work to L.F. This work was carried out with the support of the Karlsruhe Nano Micro Facility (KNMF), a Helmholtz Research Infrastructure at Karlsruhe Institute of Technology to M.S.

## Author contributions

M.M.M. and L.F. conceived the ideas and designed the work. N.L. carried out the experiments, including material preparation and characterization, device fabrication, properties, and performance measurements. E.M., Y.M.E., K.D., and M.M.M. contributed to microstructural characterization. A.N. carried out the XPS measurement and analysis. H.G. carried out XRD and DSC measurements, and M.M.M. analyzed the data. M.M.M. and A.K.V. carried out the Rietveld refinement and Transport mechanism analysis. M.I.K. performed the computational simulation analysis. J.L. contributed to the fabrication of the origami devices. M.S. assisted with the hot pressing process. T.W. carried out the S content analysis. Z.W., N.L., and M.M.M. contributed to the drawings. N.L., A.K.V., M.I.K., Z.W., J.L., A.N., and M.M.M. wrote the draft. N.L., A.K.V., U.L., and M.M.M. contributed to the discussion and editing. All authors approve the final version of the manuscript.

## Funding

## Competing interests

The authors declare no competing interests.
