## [Transparent Peer Review file · Nature Communications]

Printed origami thermoelectric generator achieves $>20 \text{ Wm}^{-2}$ from low-grade heat via material and process design

Corresponding Author: Dr Mofasser Mallick

Version 0:

Reviewer comments:

Reviewer #1

(Remarks to the Author)

This work introduced an Ag_2Se_3 based flexible material using non-stoichiometric defect engineering and sulfur doping. The demonstration of a fully printed origami-type TEG is novel and potentially relevant for IoT applications. The comments below need to be considered.

The reported device exhibits a maximum power density of 21 W m^{-2} at $\Delta T = 80 \text{ K}$, with a normalised power density of $0.3 \mu\text{W cm}^{-2} \text{ K}^{-2}$, and was claimed as record-high. However, several recent reports of flexible and printable Ag_2Se_3 -based TEGs demonstrate superior normalised power densities. For example:

<https://doi.org/10.1038/s41467-025-62336-2>

<https://doi.org/10.1016/j.nanoen.2020.105488>

<https://doi.org/10.1038/s41467-024-46183-1>

The comparison of power density in this manuscript (Fig. 1d) is primarily against the authors' own previous work, which limits the strength of the benchmarking. The authors need to explicitly compare their results with the above reports, particularly under comparable ΔT and normalisation conditions. Furthermore, the chosen $\Delta T = 80 \text{ K}$ is relatively large compared to many reports, which naturally boosts output power. The manuscript should clarify: How does the claimed performance surpass prior state-of-the-art results? Moreover, how many legs in the device module, and how does this affect scaling and comparability?

The depth of discussion on thermoelectric performance should be enhanced. What is the thermal conductivity of the fabricated film and ZT? The claimed phase transition at $T=410\text{K}$ should be evidenced. The small amount of 2% sulfur increased carrier concentration without compromising carrier mobility. What is the mechanism behind? DFT calculations are suggested to understand the band structure and therefore the performance of the printed films, particularly the impact of the sulfur doping.

Regarding Figure 4, It is unclear whether sulfur doping improves or degrades mechanical stability. Why were $x=0.05$ samples not included?

Line 221: "the mechanical behavior is similar for all samples." The $x=0$ sample showed $\sim 25\%$ resistance change after 100 bending cycles at $D = 8 \text{ mm}$, whereas the other two did not exhibit such a behaviour. This was different to the $D=6\text{mm}$ test, where $x=0.1$ showed higher resistance increase. This inconsistency requires explanation. The test of 100 bending cycles is insufficient for a device claiming flexible operation, and most flexible TEG studies report at least 1000 cycles. Extended cyclic testing is needed to prove mechanical reliability.

During practical wearable applications, when the origami device is unfolded during stretching, each leg becomes inclined relative to the hot/cold surfaces. This geometric change could influence thermal gradients and electrical performance. However, the simulations presented appear to assume perfectly vertical legs without considering such inclinations. The authors should justify this simplification.

Reviewer #2

(Remarks to the Author)

This manuscript reports on the development of a fully printed origami thermoelectric generator (TEG) using $\text{Ag}_2(\text{Se}_{1-x}\text{S}_x)_{1.05}$ -based n-type materials combined with Bi-Sb-Te p-type materials. The authors claim to achieve a record power density of 21 Wm^{-2} through sulfur substitution and non-stoichiometric defect engineering. While the topic is relevant for IoT and wearable applications, several aspects could be strengthened to enhance the scientific rigor and impact of the work.

Major Issues:

1. The core claim of this work is that sulfur substitutes into the Ag_2Se lattice. The evidence for sulfur substitution into the Ag_2Se lattice could be strengthened:
 - a. XRD shows minimal peak shifts or lattice parameter changes upon S incorporation despite claims of "contraction in lattice parameters."
 - b. The authors acknowledge that the S 2p spectrum in XPS analysis "intensity is too weak for proper quantitative analysis," undermining compositional claims.
 - c. Elemental mapping alone cannot confirm the chemical state or site occupancy of sulfur atoms. Consider providing quantitative bulk composition analysis (e.g., ICP-MS) to verify nominal compositions were achieved
 - d. Lattice parameter refinement from XRD data would help quantify structural changes.
2. The SEM images in Figures 3(d-f) could benefit from higher resolution. The description of 'relatively smooth surface' should be clarified given the visible microporosity, which may influence transport properties.
3. The transport data shows interesting optimization at $x = 0.02$. Could you elaborate on the physical mechanism behind why this composition achieves the optimal balance of carrier concentration and mobility?
4. The optimization study could be strengthened by including error bars and reproducibility data for key measurements, testing intermediate compositions (e.g., $x = 0.01$) to better define the optimum concentration, and providing sample-to-sample variation analysis.
5. The claimed phase transition at 410 K would benefit from direct structural confirmation via temperature-dependent XRD or DSC, as transport-only identification may not account for composition-dependent transition temperature shifts.
6. Thermoelectric properties after multiple heating-cooling cycles

Minor Issues

1. Figure 7 panels are missing labels and caption (f) is absent.
2. Material formulations should be standardized (e.g., consistently use $\text{Bi}_{0.5}\text{Sb}_{1.5}\text{Te}_3$).
3. Multiple instances throughout the manuscript require careful proofreading.

Reviewer #3

(Remarks to the Author)

Version 1:

Reviewer comments:

Reviewer #1

(Remarks to the Author)

The authors have addressed all my concerns and I am happy for this work to be published in Nature Communications.

Reviewer #2

(Remarks to the Author)

After carefully evaluating the newly submitted data and explanations, I find that several key scientific issues remain unresolved. The reviewer acknowledges that the authors have made a sincere effort to strengthen the manuscript by adding high-temperature XRD, DSC, reproducibility data, and extended bending-cycle tests. These additions do improve the overall quality and scope of the work. Nevertheless, the central scientific claim—that sulfur is successfully and quantitatively incorporated into the Ag_2Se lattice in a way that underpins a specific defect-engineering strategy—remains insufficiently supported by the current experimental evidence.

Given that this claim is fundamental to both the novelty and the mechanistic interpretation of the study, I do not yet consider the evidence to be at the level expected for publication in Nature Communications. I therefore recommend that the manuscript remain under major revision and that the authors be invited to address the above points with more rigorous compositional validation and appropriately calibrated mechanistic claims.

1. The authors report sulfur quantification for the $\text{Ag}_2(\text{Se}_{0.98}\text{S}_{0.02})_{1.05}$ film using carrier gas hot extraction (CGHE). However, in their own rebuttal they acknowledge that:

- a. The CGHE instrument was calibrated using organic sulfur standards, which are matrix-mismatched for inorganic chalcogenides.
- b. There is no direct evidence that lattice-bound inorganic sulfides are fully oxidized to SO_2 under the applied CGHE conditions.
- c. The presence of carbon-containing organic decomposition products may perturb the combustion/oxidation atmosphere and interfere with accurate sulfur detection.

Taken together, these statements imply that the CGHE protocol, as currently implemented, has not been validated for quantitative sulfur analysis in this specific inorganic chalcogenide matrix. Consequently, the reported sulfur content cannot yet be considered a robust basis for claims of precise sulfur incorporation into the lattice.

I therefore strongly recommend that the authors be asked to validate the sulfur content using at least one independent, matrix-appropriate method, for example inductively coupled plasma (ICP-MS or ICP-OES), electron probe microanalysis (EPMA/WDX), or a well-designed wet-chemical digestion followed by quantitative analysis. Ideally, matrix-matched inorganic sulfur standards should also be used to confirm that the CGHE calibration is valid for this class of materials. Without such validation, the defect-chemistry interpretation built on the assumed sulfur content remains on a weak foundation.

2. Insufficient Evidence that Sulfur Is Unambiguously Incorporated into the Ag₂Se Lattice

The authors have added Rietveld refinements and lattice-parameter plots to support sulfur substitution. However, the observed changes in lattice constants are extremely small and appear to be close to the plausible experimental uncertainty for laboratory XRD measurements on printed, porous films. In addition, the S 2p XPS signal remains too weak for reliable quantitative analysis, and no high-resolution STEM–EELS or equivalent atomic-scale chemical mapping has been provided. Despite this, the revised manuscript still attributes specific defect-chemistry mechanisms to sulfur substitution, such as suppression of Ag vacancies and formation of donor-type defects via excess Ag on interstitial sites. In the absence of more direct evidence, these mechanistic statements should be presented with greater caution and clearly framed as hypotheses rather than established facts.

I would therefore encourage one of the following paths: (i) provide more definitive structural and chemical-state evidence (for example, improved S 2p XPS with higher signal-to-noise, STEM–EELS line scans, or carefully error-analyzed Vegard-type trends), or (ii) revise the discussion to use more conservative language that does not overstate the level of experimental support for specific defect-chemistry scenarios.

3. Misclassification of the Printed Layer as a “Thin Film”

The reported film thickness of approximately 14–17 μm clearly places the material in the thick-film or printed-layer regime, rather than in what is typically referred to as a thin film in the thermoelectric literature (often <1 μm, and at most a few micrometres). The rebuttal does not adequately justify the continued use of the term “thin film.”

For clarity and to avoid confusion, I recommend that the authors adopt terminology such as “printed thick film,” “printed micro-thick film,” or simply “printed layer” throughout the manuscript and Supporting Information, and refrain from describing the 14–17 μm layers as thin films.

Reviewer #3

(Remarks to the Author)

Version 2:

Reviewer comments:

Reviewer #2

(Remarks to the Author)

The authors have satisfactorily addressed all major concerns raised earlier, and the manuscript is now suitable for acceptance without any further revision.

Reviewer #3

(Remarks to the Author)

KIT | LTI | Engesserstr. 13 | 76131 Karlsruhe

Phone: 721 608-45653
E-Mail: mofasser.mallick@kit.edu
Web: www.lti.kit.edu

Date: 26.10.2025

To

The Reviewers of Nature Communications

Point-by-point response to the reviewers' comments on manuscript (NCOMMS-25-57108) 'Printed origami thermoelectric generator achieves $>20 \text{ Wm}^{-2}$ from low-grade heat via material and process design'

Dear Reviewers,

We thank you for the insightful comments and suggestion, which helped us to improve the manuscript significantly. We have carried out additional measurements including, thermal analysis, temperature dependent XRD, flexibility testing, microstructural analysis, elemental analysis and reproducibility studies. We have also included more insights into the transport mechanism and revised the manuscript as per your suggestions. Please find below our point-by-point response to the comments raised by the reviewers. The reviewers' remarks are in *italics on a grey background*; our response is in regular Arial font. All the changes and modifications to the manuscript have been marked with a yellow background in one version, which we are also submitting for review. We believe that the revised version now meets the journal's requirements and we look forward to your response.

Thank you in advance.

Yours sincerely

Reply to the reviewers` comments

Reviewer: 1

Comments:

This work introduced an Ag₂Se₃ based flexible material using non-stoichiometric defect engineering and sulfur doping. The demonstration of a fully printed origami-type TEG is novel and potentially relevant for IoT applications. The comments below need to be considered.

We thank the reviewer for careful evaluation of our manuscript. We greatly appreciate the constructive feedback and are pleased to address all the comments and questions raised.

1. The reported device exhibits a maximum power density of 21 W m⁻² at ΔT = 80 K, with a normalised power density of 0.3 μW cm⁻² K⁻², and was claimed as record-high. However, several recent reports of flexible and printable Ag₂Se₃-based TEGs demonstrate superior normalised power densities. For example:

(i). <https://doi.org/10.1038/s41467-025-62336-2>

(ii). <https://doi.org/10.1016/j.nanoen.2020.105488>

(iii). <https://doi.org/10.1038/s41467-024-46183-1>

The comparison of power density in this manuscript (Fig. 1d) is primarily against the authors' own previous work, which limits the strength of the benchmarking. The authors need to explicitly compare their results with the above reports, particularly under comparable ΔT and normalisation conditions. Furthermore, the chosen ΔT = 80 K is relatively large compared to many reports, which naturally boosts output power. The manuscript should clarify: How does the claimed performance surpass prior state-of-the-art results? Moreover, how many legs in the device module, and how does this affect scaling and comparability?

We sincerely thank the reviewer for this insightful comment and for pointing out the relevant references, particularly reference (iii) (<https://doi.org/10.1038/s41467-024-46183-1>), which reports a fully printed TEG. We have now incorporated all the suggested references into the revised manuscript and updated Fig. 1 (d) to include a comprehensive comparison across reported fully/partially printed TEGs, including the suggested references. We compare the normalized power density (μWcm⁻²K⁻²) using the maximum power densities reported in the literature to ensure ΔT-independent evaluation of device performance. Additionally, to address the reviewer's concern regarding how the number of thermocouples affects scaling and comparability, we have plotted ΔT-normalized maximum power (P_{max}/ΔT²) along the x-axis. In both comparisons, our fully printed TEG demonstrates superior performance relative to the other reported fully printed devices in the context of printability and scalability.

Fig. 1 (d): Comparison of normalized power density for recently reported different partially/fully printed TEGs.

As the focus of our work is on scalable fully printed TEGs, it is important to note that all the references are not fully scalable. These studies primarily demonstrate in-plane printed TE films with thermocouples connected in series, emphasizing material-level performance while excluding the substrate thickness and parasitic heat losses. Many previously reported works lack actual details on device cross-sectional area, which is critical for accurate power density estimation. For example, in Ref (iii) (<https://doi.org/10.1038/s41467-024-46183-1>), normalized power density of 2000 nW/K²cm² is reported considering only material cross-section of 2.73E-02 mm², while the actual device application area is 1.94 mm², resulting in a true value less than order of magnitude ~28 nW/K²cm² (much less compared to our result of 300 nW/K²cm²). In contrast, our work presents a fair comparison across recent printed TEGs, as shown in Fig. 1 (d). Furthermore, the state-of-the-art printed TEGs yields limited power due to poor scalability and produce very low power (e.g. 0.8 μW for Ref (iii) at ΔT = 40 K), which is insufficient for practical applications. By contrast, our fully printed TEG achieves 200 μW under the similar conditions, representing a substantial advancement in both printability and scalability. Often times, individual leg exhibit good TE performance, this power yielding limitation arise from device dimension and number of thermocouples. Screen printing method is considered one of the scalable manufacture process facilitates large are and higher number of thermocouples which is limited in printing techniques like inkjet printing and aerogel jet printing.

2. The depth of discussion on thermoelectric performance should be enhanced. What is the thermal conductivity of the fabricated film and ZT?

We thank the reviewer for this comment. We have modified the ‘TE performance of the printed Ag₂(Se_{1-x}S_x)_{1.05} films’ section. Three additional plots have been incorporated into the Fig. 5, along with an expanded discussion of the electronic transport mechanism in the revised manuscript. The modified description is provided below.

“The temperature-dependent TE performance of the printed Ag₂(Se_{1-x}S_x)_{1.05} films for the different compositions was studied over 303 - 463 K, as shown in Fig. 5. The negative values of the Seebeck coefficient α and the Hall coefficient R_H for all samples are due to the electrons being the majority carriers. α decreases with increasing sulfur content (cf. Fig. 5a), which is due to the increase in carrier density n_H as confirmed by the Hall measurement (cf. Fig. 5d). The highest α of ~ 145-150 μV K⁻¹ is achieved at room temperature for the films with low sulfur concentrations $x \leq 0.02$. A sharp drop in α is observed at a temperature $T \approx 410$ K for all samples. This is attributed to the phase transition of Ag₂Se from an orthorhombic (beta-phase) to cubic (alpha-phase), consistent with DSC and XRD results. Fig. 5b shows the temperature-dependent electrical conductivity σ . In conjunction with the temperature-dependent carrier density, a semiconducting behaviour of all the samples is confirmed. σ increases with increasing the sulfur content up to $x = 0.02$ due to an increase in n_H (cf. Fig. 5c). It decreases for $x > 0.02$, due to the significant reduction in carrier mobility μ_H (cf. Fig. 5e). A maximum conductivity of 650 S cm⁻¹ at 300 K was observed for the printed film with $x = 0.02$. Similar to

α , σ also abruptly declines due to the phase transition at $T \approx 410$ K. The substitution of Se by S changes the defect chemistry of the Ag_2Se lattice. At low-level doping ($x = 0.02$), S incorporation likely suppresses the formation of Ag vacancies and facilitates donor-type defects through the accumulation of excess Ag on interstitial sites.³² As a result, the n_{H} increases while preserving high mobility μ_{H} (cf. Fig. 5c & e). Although the ionic radius of S^{2-} is smaller compared to Se^{2-} , lattice distortion at $x = 0.02$ is not significant enough to induce notable electron scattering. Consistently, Rietveld refinement shows a small increase in the lattice volume for $x = 0.02$, followed by a decrease at $x > 0.02$. Despite the smaller radius of S^{2-} , this slight expansion can be attributed to the presence of excess Ag on interstitial sites. However, the substitutional ionic-radius mismatch starts to dominate for $x > 0.02$, resulting in lattice contraction and higher carrier scattering.

Fig. 5. Temperature-dependent TE transport properties of n-type $\text{Ag}_2(\text{Se}_{1-x}\text{S}_x)_{1.05}$ films.

a Seebeck coefficient α , **b** Electrical conductivity σ , **c** Charge carrier concentration n_{H} ; **d** Hall coefficient R_{H} , **e** Hall mobility μ_{H} , **f** Power factor PF , **g** Ioffe-Pisarenko plot for the effective mass of 0.2289 electron mass with experimental data, **h** The weighted mobility μ_{W} , and **i** Variation of room temperature power factor and weighted mobility with S substitution.

Despite lattice contraction due to the dominant substitutional effect, the continuous rise n_{ex} for $x > 0.02$ can be attributed to the further evolution of donor-type defects. Consequently, the carrier mobility μ_{ex} decreases while retaining higher n_{ex} . The maximum carrier mobility of $1050 \text{ cm}^2 \text{ V}^{-1} \text{ s}^{-1}$ is observed at room temperature for the film with $x = 0.02$. The mobility decreases with temperature in agreement with an enhanced electron-phonon scattering.³³ Hence, the non-monotonic behavior in both the electronic transport properties and lattice parameters arises from the interplay between the vacancy suppression, lattice expansion and subsequent lattice distortion, facilitating an optimum TE power factor $PF(\alpha^2 \sigma)$ at $x = 0.02$. It is plotted as a function of temperature as shown in Fig. 5f. The maximum power factor of $\sim 16 \mu\text{W cm}^{-1} \text{ K}^2$ is obtained for the film with $x = 0.02$ at 360 K, indicating that the inclusion of a small fraction of sulfur facilitates the optimum TE performance for $\text{Ag}_2(\text{Se}_{1-x}\text{S}_x)$. The power factor is enhanced by 40 % for $x = 0.02$ compared to the pristine film. Further, at higher temperatures, the power factor also reflects the phase transition. Thermal conductivity is estimated to be $0.59 \text{ Wm}^{-1} \text{ K}^{-1}$ for the $x = 0.02$ film based on the measured value of a non-pressed film. The predicted ZT value of the film for $x = 0.02$ is 0.7 at RT (see supporting information note S4). To gain deeper insight into the transport mechanism, weighted mobility (μ_{w}) and a theoretical Ioffe-Pisarenko curve have been evaluated using the single-parabolic band (SPB) model. The temperature-dependent weighted mobility μ_{w} was calculated from the electrical conductivity and Seebeck coefficient using the following equation;³³

$$\mu_{\text{w}} = \frac{3 h^3 \sigma}{8 \pi e (2 m_e k_B T)^{3/2}} \left[\frac{\exp\left[\frac{|\alpha|}{k_B/e} - 2\right]}{1 + \exp\left[-S\left(\frac{|\alpha|}{k_B/e} - 1\right)\right]} + \frac{\frac{S}{\pi^2} \frac{|\alpha|}{k_B/e}}{1 + \exp\left[S\left(\frac{|\alpha|}{k_B/e} - 1\right)\right]} \right]$$

In addition, the theoretical Ioffe-Pisarenko curve has been calculated by the SPB model with acoustic phonon scattering for an effective mass of $0.2289 m_0$ using the following relation^{34,35}

$$\alpha : : \frac{k_B}{\sigma} \left(\frac{2k_B^3(\eta)}{k_B^3(\eta)} \right) \quad (2)$$

where the Fermi integral of order j is defined as:

$$F_j(\eta) = \int_0^\infty \frac{\epsilon^j d\epsilon}{1 + \exp(\epsilon - \eta)} \quad (3)$$

The chemical carrier concentration is expressed as:

$$n : : A_{\text{eff}} \left(\frac{2m^* k_B T}{\hbar^2} \right)^{3/2} F_{3/2}(\eta) \quad (4)$$

while the Hall carrier concentration is given by:

$$n_H : : \frac{n}{r_H} \quad (5)$$

with the Hall factor

$$r_{H2} = \frac{3}{2} \frac{F_{1/2}(\eta)}{F_{3/2}(\eta)} \frac{e^{-\epsilon/\eta}}{2k_B^* T} \quad (6)$$

In these expressions, η is the reduced chemical potential, ϵ is the reduced energy, k_B is the Boltzmann constant, e is the bare electronic charge, m_n^* is the density-of-states effective mass, T is the temperature, and h is the Planck constant. Similar trend is also observed in the temperature dependent μ_{WV} for all films (cf. Fig. 5h). The μ_{WV} is one of the key descriptors of electronic transport properties, comprises of μ_{CX} and effective mass (m^*). It is found to be the highest at $x = 0.02$ for the beta- Ag_2Se phase, indicating the optimum trade-off between μ_{CX} and r_{CX} . The μ_{WV} decreases rapidly due to carrier scattering for $x > 0.02$, despite increasing of r_{CX} . To further understand the defect chemistry of the Ag_2Se lattice upon S substitution, the theoretical Ioffe-Pisarenko curve is shown in Fig. 5g. It shows that the experimental effective masses in all samples lie around the simulated curve, with little deviation found upon high S substitution due to the point defects. Fig. 5i shows the room temperature x -dependent power factor and μ_{WV} , and confirm that $x = 0.02$ is the optimum S substitution for power generation for low temperature TE devices. To check the reproducibility, two printed $\text{Ag}_2(\text{Se}_{0.98}\text{S}_{0.02})_{1.05}$ films were prepared from two different batches of inks. The TE performance of films is calculated and compared in Supplementary Fig. S5. It is found that the overall power factor value remains within 10 %. The TE performance of the printed $\text{Ag}_2(\text{Se}_{0.98}\text{S}_{0.02})_{1.05}$ film was also measured in 5 heating/cooling cycles to evaluate the thermal stability (cf. Supplementary Fig. S5). There is a small hysteresis found between heating and cooling. However, the results show the film is thermally stable and produces similar performance. The TE properties of the printed p-type $\text{Bi}_{0.5}\text{Sb}_{1.5}\text{Te}_3$ film were also studied and agreed with previous studies. The data is included in the supporting information (cf. Fig. S7)."

The thermal conductivity of non-pressed printed film for $x=0.02$ was measured using Linseis thin film analyser and the same is estimated for compacted printed film. The following discussion is included in the supporting information.

Then the thermal conductivity was estimated for the $x = 0.02$ composition based on the measured value of a non-pressed film, in order to predict its ZT at room temperature using the following relation (see supplementary note S4).

$$\kappa_{eff} = \kappa_{bulk}(1 - P)$$

Where κ_{eff} and κ_{bulk} are the thermal conductivity of the printed film and bulk Ag_2Se . P is the porosity. The porosity of the non-pressed and pressed film for $x=0.02$ estimated to be $P_{60} = 60\%$ and

$P_{45} = 45\%$ respectively. The estimated thermal conductivity of the pressed printed film $\kappa_{eff,49}$ for $x=0.02$ can be calculated using the following expression;

$$\kappa_{eff,45} = \kappa_{eff,60} \frac{(1 - P_{45})}{(1 - P_{60})}$$

Thermal conductivity of the non-pressed film $\kappa_{eff,60}$ ($0.43 \text{ Wm}^{-1}\text{K}^{-1}$) is measured by Linseis thin film analyser. Hence, $\kappa_{eff,45} = 0.59 \text{ Wm}^{-1}\text{K}^{-1}$ at room temperature and corresponding $ZT=0.7$.

However, the estimated value may not be error free. Measuring true thermal conductivity of our compacted printed film remains a challenge. In general, we employ two well-known setups to measure the thermal conductivity of printed film; a) Thin film analyser (Linseis thin film analyser (TFA) system) and b) time domain reflection (TDR) measurement setup. The Linseis TFA was developed by V. Linseis et. al. [J. Mater. Res., 2016, 31, 20]. The TFA chip is fabricated on a silicon wafer where two thin heater with a width $< 5 \mu\text{m}$ are deposited on free standing Si_3N_4 membranes of 100 nm which is surrounded by an Au rim. The heaters are connected in a 4-wire configuration. However, the membrane does not withstand even a slight pressure, hence we could not measure the thermal conductivity using the TFA.

We have also tried to employ TDR measurement setup to determine the thermal conductivity. Unfortunately, it requires a light reflecting surface of a measuring film. As the Ag_2Se film is thin ($10 \mu\text{m}$), therefore not suitable for polishing, and blackish, hence getting a light reflecting surface was not possible. Therefore, we were also not able to measure the thermal conductivity using TDR measurement setup. This remains a challenge which will take dedicated efforts in future projects. While we agree that a direct measurement of the thermal conductivity is desirable, we do not see a short-term solution for a reliable measurement. We hope to solve the current problem in the future.

3. The claimed phase transition at $T=410\text{K}$ should be evidenced. The small amount of 2% sulfure increased carrier concentration without compromising carrier mobility. What is the mechanism behind? DFT calculations are suggested to understand the band structure and therefore the performance of the printed films, particularly the impact of the sulfure doping.

We thank the reviewer for the suggestion. We have now performed high-temperature XRD and DSC measurements and replaced Fig. 3 with the new data. The data clearly supports the notion of a phase transition at 410 K. The revised description of the '*Thermal and phase analysis*' section is provided below.

“To evaluate phase purity and structural changes upon sulfur substitution, X-ray diffraction (XRD) was carried out on pristine non-stoichiometric $\text{Ag}_2\text{Se}_{1.05}$ and sulfur-substituted samples with nominal compositions $\text{Ag}_2(\text{Se}_{1-x}\text{S}_x)_{1.05}$ ($x = 0, 0.02, 0.05, \text{ and } 0.1$), as presented in Supplementary Fig. S1 and Fig. 2. Rietveld refinement analysis of all the XRD patterns indicates that the films are crystallized in the orthorhombic phase of beta- Ag_2Se , in agreement with the standard JCPDS reference (No. 24–1041).³⁰ The diffraction peaks can be indexed to the orthorhombic structure with space group symmetry $\text{P}2_12_12_1$, with no significant secondary phases, indicating high phase purity and crystallinity in

all samples. Although the printed films are $\text{Se}_{1-x}\text{S}_x$ -rich, their XRD patterns correspond to the stoichiometric beta- Ag_2Se phase, consistent with a previous report.²⁴

Fig. 1. Thermal and crystallographic analysis of printed $\text{Ag}_2(\text{Se}_{1-x}\text{S}_x)_{1.05}$ films.

a DSC heating and cooling cycle of the printed film with $x = 0.02$. **b** Temperature-dependent XRD patterns of the printed $\text{Ag}_2(\text{Se}_{0.98}\text{S}_{0.02})_{1.05}$ film. **c** Rietveld refinement of XRD patterns of the beta- Ag_2Se (at 303 K) and alpha- Ag_2Se phase (at 433 K). **d-g** The variation of lattice parameters with sulfur substitutions.

To analyze the structural change of the orthorhombic beta- Ag_2Se phase with temperature, differential scanning calorimetry (DSC) and high-temperature XRD studies were performed (cf. Fig. 2a-c). Both measurements reveal that the room temperature (RT) orthorhombic beta- Ag_2Se phase undergoes a structural transition to cubic alpha- Ag_2Se phase with space group $Im\bar{3}m$ at approximately $T = 410$ K.

An endothermic peak at $T = 410$ K during heating and an exothermic peak at $T = 368$ K during cooling, are observed in the DSC cycle, which indicates these reversible phase transitions of Ag_2Se . The Rietveld refinements of the XRD patterns collected at RT and at 433 K further confirm the phase transition and provide insight into the crystal structures (cf. Fig. 2c). In the beta- Ag_2Se lattice, both Ag and Se atoms occupy Wyckoff position 4a (x,y,z), whereas in alpha- Ag_2Se they are located in two inequivalent positions 12d ($1/4, 1/2, 0$) and 2a ($0, 0, 0$), respectively. The lattice parameters of the RT beta- Ag_2Se lattice are plotted for different ‘S’ contents in Fig. 2d-g. A slight lattice expansion is observed for $x = 0.02$, followed by contraction for $x > 0.02$. This behaviour can be attributed to the suppression of Ag-vacancy formation and the subsequent migration of excess Ag into interstitial sites. However, at high doping levels ($x > 0.02$), the substitutional effect of smaller S^{2-} ions dominates, resulting in lattice contraction. Consistently, this structural evolution correlates with the trends in the transport properties of the S-doped Ag_2Se printed films.”

The detailed transport mechanism has been included in the revised manuscript (please see the reply 2). We agree with the reviewer that first-principles (DFT) calculations would provide deeper insights into the electronic band structure. While not being specialists in quantum chemical calculations, we have already attempted to carry out DFT calculations of our materials. There are, however, several challenges in doing so. Firstly, even the fully crystallized structure itself is rather complicated as confirmed from discussions with our colleagues in quantum chemistry. Secondly, even after successful DFT calculations, there is no straightforward prediction of macroscopic properties such as charge carrier mobilities for our microcrystalline and porous materials. Our printed TE films are less dense (71 % of theoretical value) and contain a large density of grain boundaries, and a fairly large porosity, making transport behavior more complicated than in the purely crystalline bulk material. As a result, e.g., the bulk electrical conductivity of Ag_2Se is significantly still higher than the printed counterpart. The TE-performance nevertheless of the printed TE film can still be as good as bulk as the thermal conductivity is also reduced. We will continue to explore a more rigid theoretical treatment, however, such detailed computational investigations are well beyond the scope of the present work.

4. Regarding Fig. 4, It is unclear whether sulfur doping improves or degrades mechanical stability. Why were $x=0.05$ samples not included? Line 221: “the mechanical behavior is similar for all samples.” The $x=0$ sample showed ~25% resistance change after 100 bending cycles at $D = 8$ mm, whereas the other two did not exhibit such a behaviour. This was different to the $D=6$ mm test, where $x=0.1$ showed higher resistance increase. This inconsistency requires explanation. The test of 100 bending cycles is insufficient for a device claiming flexible operation, and most flexible TEG studies report at least 1000 cycles. Extended cyclic testing is needed to prove mechanical reliability.

We agree that 100 bending cycles is not sufficient to test mechanical reliability. Therefore, we have performed cyclic bending test for 1000 cycles on all the films at $D=10.8$ mm. The results are now included in the Fig. 4 of the revised manuscript. The following Figure and discussion have been included in the revised manuscript.

Fig. 4: Mechanical properties of n-type $\text{Ag}_2(\text{Se}_{1-x}\text{S}_x)_{1.05}$ films. e) Normalized resistance of the printed films for $D=8\text{mm}$ during 1000 bending cycles. f) variation of standard deviation in normalized resistance data with ‘x’ during 1000 bending cycles.

Furthermore, bending tests were carried out for 1000 cycles on all films at a $D=10.8\text{ mm}$ to assess their mechanical tolerance (cf. Fig. 4 (e) & (f)). The normalized resistance of the undoped Ag_2Se film increases abruptly after 800 bending cycles due to crack formation, whereas the printed films with $x>0$ maintain stable resistance up to 1000 bending cycles (cf. Fig. 4 (e)). To quantify the effect of ‘S’ substitution on mechanical flexibility, the standard deviation (SD) of normalized resistance is plotted as a function of ‘x’ (cf. Fig. 4 (f)). The SD decrease with increasing ‘x’, indicating improved ductility. The SD values vary between 1 to 0.40 for $x>0.02$ films. These results confirm the excellent durability of the S-doped printed films under repeated bending. We conclude that the material with $x=0.02$ marks a good compromise in simultaneously optimizing the electronic and mechanical properties.

5. During practical wearable applications, when the origami device is unfolded during stretching, each leg becomes inclined relative to the hot/cold surfaces. This geometric change could influence thermal gradients and electrical performance. However, the simulations presented appear to assume perfectly vertical legs without considering such inclinations. The authors should justify this simplification.

We agree with the reviewer that TE legs will be inclined relative to the hot and cold surfaces during unfolding. While in some applications where high power density is not required, unfolding of the origami TEG may work sufficiently, however the primary objective of our work is that all the TE legs remain folded during operation. For larger areas and curved surfaces we intend to add more thermocouples. In real-world applications such as wearables or IoT devices, the origami TEG would not be unfolded or stretched. Instead, multiple folded TEG units would be interconnected to form a band or cube-like structure (see below Fig.), ensuring that the legs remain vertically aligned between the hot and cold sides (potential prototypes being developed by our research group are shown in the

Fig.). The compression or force on the hot or cold side would be minimal and should not cause the legs to stretch or tilt toward either side.

Folded origami TEG prototypes.

Reviewer. 2

Comments

This manuscript reports on the development of a fully printed origami thermoelectric generator (TEG) using $\text{Ag}_2(\text{Se}_{1-x}\text{S}_x)_{1.05}$ -based n-type materials combined with Bi-Sb-Te p-type materials. The authors claim to achieve a record power density of 21 Wm^{-2} through sulfur substitution and non-stoichiometric defect engineering. While the topic is relevant for IoT and wearable applications, several aspects could be strengthened to enhance the scientific rigor and impact of the work.

We are grateful to the reviewer for the insightful feedback and comments, which helped us to improve the quality of our manuscript. We have revised the manuscript according to the reviewer's comments and suggestions, and detailed responses are provided below.

Major Issues:

1. The core claim of this work is that sulfur substitutes into the Ag_2Se lattice. The evidence for sulfur substitution into the Ag_2Se lattice could be strengthened:

a. XRD shows minimal peak shifts or lattice parameter changes upon S incorporation despite claims of "contraction in lattice parameters."

b. The authors acknowledge that the S 2p spectrum in XPS analysis "intensity is too weak for proper quantitative analysis," undermining compositional claims.

c. Elemental mapping alone cannot confirm the chemical state or site occupancy of sulfur atoms. Consider providing quantitative bulk composition analysis (e.g., ICP-MS) to verify nominal compositions were achieved

d. Lattice parameter refinement from XRD data would help quantify structural changes.

We agree with the reviewer that evidence of sulfur substitution should be improved. In the line we have revised the manuscript to clarify the following:

(a) & (d): We have now performed high-temperature XRD and DSC measurements and replaced Fig. 3 with the new data. We have now performed Rietveld refinement of the room-temperature XRD patterns for all samples (see Fig. S1). The corresponding lattice parameters for different sulfur contents (x) are plotted in Fig. 2 (d–f). The detailed description of the 'Thermal and phase analysis'

section has been included in the updated manuscript and in our response to Comment 2 of Reviewer 1.

Fig. S1: Rietveld refinement of the room temperature XRD patterns for the $Ag_2(Se_{1-x}S_x)_{1.05}$ thin films

(b) & (c): We have now performed carrier gas hot extraction (CGHE) analysis to quantify the ‘S’ content. The following discussion has been included in the revised manuscript.

To quantify the amount of ‘S’ in the printed $Ag_2(Se_{0.98}S_{0.02})_{1.05}$ film, carrier gas hot extraction (CGHE) analysis was performed. The compound was melted under a carrier gas and the existing ‘S’ react to form SO_2 . The quantitative determination of three samples is then done using selective infrared detection. The ‘S’ content was measured to be 0.47 at. % with a limit of quantification of 0.0143 at %, a standard deviation of 0.0030, and a measurement uncertainty of ± 0.0383 . The measured ‘S’ content is slightly lower than the theoretical value, which could be attributed to the measurement limitations of the CGHE method for inorganic chalcogenide film. Here, the analyzer was calibrated using organic ‘S’ standards, which could underestimate the quantification. In addition, a noticeable carbon signal was also detected in the sample due to decomposition of organic ingredients. This may have affected the measurement by altering the combustion atmosphere.

2. The SEM images in Fig.s 3(d-f) could benefit from higher resolution. The description of 'relatively smooth surface' should be clarified given the visible microporosity, which may influence transport properties.

We thank the reviewer for the suggestion. Higher-magnification SEM images have now been added in the revised manuscript Fig. 4 (d-f).

Fig. 3. SEM microstructures of sintered and pressed $\text{Ag}_2(\text{Se}_{1-x}\text{S}_x)_{1.05}$ thin film (d) $x=0$ (e) $x=0.02$ (f) $x=0.1$.

3. The transport data shows interesting optimization at $x = 0.02$. Could you elaborate on the physical mechanism behind why this composition achieves the optimal balance of carrier concentration and mobility?

We thank the reviewer for highlighting this important point. A detailed discussion has now been included in the revised manuscript. We kindly refer the reviewer to our response to Comment 2 of Reviewer 1 for further details.

4. The optimization study could be strengthened by including error bars and reproducibility data for key measurements, testing intermediate compositions (e.g., $x = 0.01$) to better define the optimum concentration, and providing sample-to-sample variation analysis.

We have re-synthesized the sample and performed TE transport measurements, with the reproducibility data provided in the Supporting Information (Fig. S5). The sample-to-sample variation in the power factor for the film with $x = 0.02$ was within $\pm 10\%$, demonstrating good reproducibility. Additionally, a sample with $x = 0.01$ was prepared, and the corresponding results are included in the Supporting Information (Fig. S6). The performance of the $x = 0.01$ composition is found to lie between those of the $x = 0$ and $x = 0.02$ samples.

Fig. S5: Comparison of TE properties of n-type printed films for 'x=0.02' prepared from two different batches of inks (I and II): (a) Seebeck coefficient α (b) Electrical conductivity σ and (c)

Fig. S6: TE properties of n-type printed films for 'x=0.01' prepared film: (a) Seebeck coefficient α (b) Electrical conductivity σ and (c) Power factor $\alpha^2\sigma$.

5. The claimed phase transition at 410 K would benefit from direct structural confirmation via temperature-dependent XRD or DSC, as transport-only identification may not account for composition-dependent transition temperature shifts.

We thank the reviewer for pointing out this important issue. We have now performed high-temperature XRD and DSC measurements and replaced Fig. 3 with the new data. Please see the reply to the comment 3 of reviewer 1.

6. Thermoelectric properties after multiple heating-cooling cycles

We have now performed cyclic heating-cooling tests up to 5 cycles between 300 K and 460 K for optimized composition x=0.02. The transport properties (α and σ) remain stable within $\pm 10\%$ variation, indicating good thermal stability of the printed films. The new data have been added to the Supporting Information (Fig. S5).

The TE performance of the printed $\text{Ag}_2(\text{Se}_{0.98}\text{S}_{0.02})_{1.05}$ film was also measured in 5 heating/cooling cycles. There is small hysteresis found between heating and cooling. However, the results show the film thermally stable and produces a similar performance.

Fig. S5: TE properties of the printed $\text{Ag}_2(\text{Se}_{0.98}\text{S}_{0.02})_{1.05}$ film during 5 heating/cooling cycles.

Minor Issues

1. Fig. 7 panels are missing labels and caption (f) is absent.

We thank the reviewer for finding these missing levels and the caption of Fig. 7(f). We have corrected this information in Fig. 7 of the revised manuscript.

2. Material formulations should be standardized (e.g., consistently use $\text{Bi}_{0.5}\text{Sb}_{1.5}\text{Te}_3$).

We thank the reviewer for finding out the mistakes. We have thoroughly checked the manuscript and standardized the notation throughout for consistency.

3. Multiple instances throughout the manuscript require careful proofreading.

We have carefully re-read the whole manuscript. Several grammatical errors and inconsistencies have been corrected in the revised manuscript.

Reviewer. 3

We thank Reviewers for their time and thoughtful co-review. We greatly appreciate your contribution to improving the quality of our manuscript.

Finally, we would like to thank the esteemed Editor and his team for their fruitful instructions and suggestions, which have significantly improved the scientific and non-scientific content of the manuscript.

KIT | LTI | Engesserstr. 13 | 76131 Karlsruhe

Phone: 721 608-45653
E-Mail: mofasser.mallick@kit.edu
Web: www.lti.kit.edu

Date: 14.12.2025

To

The Reviewers of Nature Communications

Point-by-point response to the reviewers' comments on manuscript (NCOMMS-25-57108A) 'Printed origami thermoelectric generator achieves $>20 \text{ Wm}^{-2}$ from low-grade heat via material and process design'

Dear Reviewers,

We thank you again for the insightful and constructive comments provided on our manuscript. We have carefully addressed the comments and performed additional measurements including EPMA-WDX and synchrotron-based XPS at the HESGM beamline of BESSY II (Helmholtz-Zentrum Berlin), followed by revision of the manuscript in accordance with the suggestions raised. Please find below our point-by-point response to the comments raised by the reviewers. The reviewers' remarks are in *italics on a grey background*; our response is in regular Arial font. All the changes and modifications to the manuscript have been marked with a yellow background in one version, which we are also submitting for review. We believe that the revised version now meets the journal's requirements and we look forward to your response.

Thank you in advance.

Yours sincerely

Reply to the reviewers' comments

Reviewer: 1

Comments:

The authors have addressed all my concerns and I am happy for this work to be published in Nature Communications.

We appreciate the reviewer for the positive evaluation of our manuscript and recommendation for publication.

Reviewer. 2

Comments

After carefully evaluating the newly submitted data and explanations, I find that several key scientific issues remain unresolved. The reviewer acknowledges that the authors have made a sincere effort to strengthen the manuscript by adding high-temperature XRD, DSC, reproducibility data, and extended bending-cycle tests. These additions do improve the overall quality and scope of the work. Nevertheless, the central scientific claim—that sulfur is successfully and quantitatively incorporated into the Ag_2Se lattice in a way that underpins a specific defect-engineering strategy—remains insufficiently supported by the current experimental evidence.

Given that this claim is fundamental to both the novelty and the mechanistic interpretation of the study, I do not yet consider the evidence to be at the level expected for publication in Nature Communications. I therefore recommend that the manuscript remain under major revision and that the authors be invited to address the above points with more rigorous compositional validation and appropriately calibrated mechanistic claims.

1. The authors report sulfur quantification for the $\text{Ag}_2(\text{Se}_{0.98}\text{S}_{0.02})_{1.05}$ film using carrier gas hot extraction (CGHE). However, in their own rebuttal they acknowledge that:

a. The CGHE instrument was calibrated using organic sulfur standards, which are matrix-mismatched for inorganic chalcogenides.

b. There is no direct evidence that lattice-bound inorganic sulfides are fully oxidized to SO_2 under the applied CGHE conditions.

c. The presence of carbon-containing organic decomposition products may perturb the combustion/oxidation atmosphere and interfere with accurate sulfur detection.

Taken together, these statements imply that the CGHE protocol, as currently implemented, has not been validated for quantitative sulfur analysis in this specific inorganic chalcogenide matrix. Consequently, the reported sulfur content cannot yet be considered a robust basis for claims of precise sulfur incorporation into the lattice.

I therefore strongly recommend that the authors be asked to validate the sulfur content using at least one independent, matrix-appropriate method, for example inductively coupled plasma (ICP-MS or ICP-OES), electron probe microanalysis (EPMA/WDX), or a well-designed wet-chemical digestion followed by quantitative analysis. Ideally, matrix-matched inorganic sulfur standards should also be used to confirm that the CGHE calibration is valid for this class of materials. Without such validation, the defect-chemistry interpretation built on the assumed sulfur content remains on a weak foundation.

We thank the reviewer for highlighting the shortcomings of the initial characterization method used to determine the sulfur concentration. We acknowledge that quantitative validation of sulfur incorpo-

ration is vital to the scientific foundation of our defect-chemistry interpretation. As per reviewer's suggestion, we have now performed additional wavelength-dispersive electron probe microanalysis (EPMA-WDX), and the corresponding results and discussion have now been included in the revised manuscript.

The following description has been added to the Experimental Section.

'Electron-probe microanalyses (EPMA) of the printed $\text{Ag}_2(\text{Se}_{0.98}\text{S}_{0.02})_{1.05}$ film were carried out using a JEOL JXA-8200 wavelength-dispersive electron microprobe. Analyses were performed at an accelerating voltage of 15 kV, a beam current of 10 nA, using a focused beam diameter of 1 μm . Peak and background counting times were optimized for low-level detection of chalcophile elements (50 s on-peak, 25 s background). Matrix corrections were applied using the $\phi(\rho z)$ routine implemented in the JEOL software. Well-characterized standards were used for elemental calibrations, i.e. Galena (PbS) for sulfur as well as synthetic metallic Ag for silver, and synthetic metallic Se for selenium analysis. Detection limits under these analytical conditions were generally below c. 300 ppm for Ag and Se and below c. 100 ppm for S. Analyses were monitored for potential beam damage and element loss; none was detected under the chosen beam conditions. Repeated measurements of standards and internal reference materials yielded analytical reproducibility better than 1% relative for major elements and 5% relative for minor elements.'

The following description has been incorporated in the section '*Chemical, microstructural and elemental analyses of printed $\text{Ag}_2(\text{Se}_{1-x}\text{S}_x)_{1.05}$ films*'.

Furthermore, the elemental composition of the printed $\text{Ag}_2(\text{Se}_{0.98}\text{S}_{0.02})_{1.05}$ film was investigated by wavelength-dispersive X-ray spectroscopy (WDX) using electron probe microanalysis (EPMA) at four different probe points (cf. Fig. S3 (b) and Table S1). The quantitative results, reported in weight percent (wt. %) and atomic percent (at. %), confirm a consistent distribution of Ag, Se, and S across the probe points. The Ag amount varies between 65.01 and 67.86 at. %, Se between 31.45 and 34.37 at. %, and S between 0.53 and 0.76 at%. The presence of sulfur indicates a reasonably uniform distribution at the micrometre scale. The small compositional variations are probably due to film porosity and matrix effects in the printed film. The nitrogen-free (NF) value of 0.03 at all probe points indicates minor background contributions. In addition, a negligible trace of Se-rich phase is observed in the printed film (light grey region in Fig. S3 (b)). To provide a statistical overview of all elemental compositions, the mean values and standard deviations (SD) from the four probe points are calculated and presented in Table 1. The composition amounts to Ag: 68.12 ± 3.29 wt. % (66.64 ± 1.19 at.

%), Se: 24.51 ± 2.12 wt. % (32.70 ± 1.22 at. %), and S: 0.200 ± 0.022 wt. % (0.65 ± 0.10 at. %). The presence of Ag, Se and S in the printed film are in good agreement with the nominal composition of $\text{Ag}_2(\text{Se}_{0.98}\text{S}_{0.02})_{1.05}$, confirming an effective sulfur incorporation within the printed material. The total measured Ag, Se and S content accounts to 92.82 ± 5.11 wt. %, which is slightly below 100 wt. %. This deviation is attributed to the presence of porosity and residual organic ingredients in the printed film, which are not detected by EPMA.

Table 1. EPMA-WDX analysis of Ag, Se, and S in printed $\text{Ag}_2(\text{Se}_{0.98}\text{S}_{0.02})_{1.05}$ film.

Element	wt. % (mean \pm SD)	at. % (mean \pm SD)
Ag	68.12 ± 3.29	66.64 ± 1.19
Se	24.51 ± 2.12	32.70 ± 1.22
S	0.20 ± 0.022	0.65 ± 0.10
Total	92.82 ± 5.11	100.00 ± 0.00

2. Insufficient Evidence that Sulfur Is Unambiguously Incorporated into the Ag_2Se Lattice

The authors have added Rietveld refinements and lattice-parameter plots to support sulfur substitution. However, the observed changes in lattice constants are extremely small and appear to be close to the plausible experimental uncertainty for laboratory XRD measurements on printed, porous films. In addition, the S 2p XPS signal remains too weak for reliable quantitative analysis, and no high-resolution STEM–EELS or equivalent atomic-scale chemical mapping has been provided.

Despite this, the revised manuscript still attributes specific defect-chemistry mechanisms to sulfur substitution, such as suppression of Ag vacancies and formation of donor-type defects via excess Ag on interstitial sites. In the absence of more direct evidence, these mechanistic statements should be presented with greater caution and clearly framed as hypotheses rather than established facts.

I would therefore encourage one of the following paths: (i) provide more definitive structural and chemical-state evidence (for example, improved S 2p XPS with higher signal-to-noise, STEM–EELS line scans, or carefully error-analyzed Vegard-type trends), or (ii) revise the discussion to use more conservative language that does not overstate the level of experimental support for specific defect-chemistry scenarios.

We thank the reviewer for this technically detailed assessment. We acknowledge that explicit identification of sulfur site occupancy and the associated defect mechanism is challenging, particularly at low sulfur concentrations and in printed, porous films. In response to the reviewer's comments, we have taken both suggested approaches: (i) we have added more key chemical-state evidence using synchrotron-based XPS, and (ii) we have rephrased the sentences in the revised the manuscript to ensure that all defect-chemistry interpretations are explicitly framed as hypothesis-driven rather than as established facts.

To provide more insight into the chemical-state, we have performed high-resolution synchrotron-based XPS measurements at the HESGM beamline of BESSY II (Helmholtz-Zentrum Berlin). Detailed discussion on Ag3d, Se3d, and Se3p/S2p spectra have been included in the revised manuscript (see below). While synchrotron-based XPS facilitates considerably improved chemical-state sensitivity compared to the previous lab XPS, it still does not enable unambiguous determination of

bulk sulfur site occupancy. Hence, as per the reviewer's recommendation, we have revised the manuscript rephrasing the relevant sentences to adopt a more conservative interpretation of the defect chemistry. Statements implying definitive suppression of Ag vacancies or formation of donor-type defects via interstitial Ag have been softened and are now presented as physically plausible scenarios consistent with the observed experimental data. We acknowledge that direct confirmation of sulfur site occupancy and defect configurations would require atomic-scale techniques such as STEM–EELS, which are beyond the scope of the present study.

Chemical, microstructural and elemental analyses of printed $\text{Ag}_2(\text{Se}_{1-x}\text{S}_x)_{1.05}$ films

Firstly, the survey XP spectra was measured and peaks corresponding to oxygen (O1s), carbon (C1s), silver (Ag3d) and selenium (Se3p/Se3d) could be recognized. Since the amount of sulfur is small, corresponding peaks are not recognized in the survey XP spectra. Then, the detailed Ag3d, Se3d, and Se3p/S2p XP spectra were recorded. In the latest case the measurements were carried out using synchrotron-based XPS at HESGM beamline of synchrotron facility BESSY II (Helmholtz-Zentrum Berlin, Berlin). The detailed Ag3d spectrum (cf. Fig. 3a) consists of one doublet with a fixed splitting of 6 eV, keeping the intensity ratio of 3:2 between Ag 3d_{5/2} and Ag 3d_{3/2} lines. The position of Ag 3d_{3/2} lines at 367.9 eV is in good agreement with the Ag⁺ oxidation state, which is typical for Ag₂Se.^{8,15,31} A very small additional component (corresponding to Ag⁰ state) at 368.5 eV has also been observed in the measured spectrum. Since its intensity is on the level of experimental errors, the variation of the area parameters in the range between 2 and 10 % during the fitting procedure does not result in any visible changes in the envelope spectra. The detailed Se3d spectrum (cf. Fig. 3 b) was deconvoluted with two Se3d doublets, applying the splitting of 0.86 eV, intensity ratio of 3:2 in both cases. The binding energies of the Se 3d_{5/2} components at 53.4 eV and 55.9 eV are in good agreement with previous experiments, corresponding to Se²⁻ and Se⁰ oxidation states.^{8,15} From the intensity comparison of both components (~ 12:1) the contribution of Se⁰ of 7 % is estimated. The detailed Se3p/S2p spectrum is presented in Fig. S3 (a) (open symbols). The shape of this spectrum is rather complicate, but its analysis could be partially simplified using Se²⁻/Se⁰ parameters obtained from Se3d spectrum (cf. Fig 3 b). The corresponding Se3p components are presented with green (Se²⁻) and blue (Se⁰) curves. The rest of the intensity could be fitted with three S2p doublets corresponding to S²⁻/S⁰/S^{δ+} states, which are presented in light blue, magenta and orange curves. The presence of S²⁻ component results from a partial substitution of selenium atoms with sulfur. The S2p²⁻/Se3p²⁻ intensity ratio is about 5 %, which is in a good agreement with the expected degree of sulfur substitution. The presence of two other components (neutral S⁰ and oxidized S^{δ+}) was unexpected and could be explained with a co-existence of unreacted sulfur layer on the $\text{Ag}_2(\text{Se}_{1-x}\text{S}_x)_{1.05}$

film. In order to check this supposition more surface-sensitive XPS measurements were repeated with an excitation energy of photons of 300 eV. For these conditions the signal was collected from the thickness of 1-2 nm only. The corresponding Se3p/S2p spectrum is shown at inset in Fig. S3 (a). The intensity of “bulk” Se3p_{3/2} component is drastically decreased, but S2p⁰/S2p^{δ+} peaks are still clearly visible, confirming our supposition.’

3. Misclassification of the Printed Layer as a “Thin Film”

The reported film thickness of approximately 14–17 μm clearly places the material in the thick-film or printed-layer regime, rather than in what is typically referred to as a thin film in the thermoelectric literature (often <1 μm, and at most a few micrometres). The rebuttal does not adequately justify the continued use of the term “thin film.”

For clarity and to avoid confusion, I recommend that the authors adopt terminology such as “printed thick film,” “printed micro-thick film,” or simply “printed layer” throughout the manuscript and Supporting Information, and refrain from describing the 14–17 μm layers as thin films.

We apologize for the oversight and thank the reviewer for this valuable comment. We have now corrected the terminology throughout the revised manuscript.

Reviewer. 3

We thank Reviewers for their time and thoughtful co-review. We greatly appreciate your contribution to improving the quality of our manuscript.